# DISTRIBUTED EXTRA-GRADIENT WITH OPTIMAL COMPLEXITY AND COMMUNICATION GUARANTEES

**Ali Ramezani-Kebrya**[*†] **Kimon Antonakopoulos**[*‡] **Igor Krawczuk**[*‡] **Justin Deschenaux**[*‡]
**Volkan Cevher**[‡]
[†]`ali@uio.no`    [‡]`firstname.lastname@epfl.ch`

## ABSTRACT

We consider monotone variational inequality (VI) problems in *multi-GPU* settings where multiple processors/workers/clients have access to local *stochastic dual vectors*. This setting includes a broad range of important problems from distributed convex minimization to min-max and games. Extra-gradient, which is a de facto algorithm for monotone VI problems, has not been designed to be communication-efficient. To this end, we propose a quantized generalized extra-gradient (Q-GenX), which is an unbiased and adaptive compression method tailored to solve VIs. We provide an adaptive step-size rule, which adapts to the respective noise profiles at hand and achieve a fast rate of $\mathcal{O}(1/T)$ under relative noise, and an order-optimal $\mathcal{O}(1/\sqrt{T})$ under absolute noise and show distributed training accelerates convergence. Finally, we validate our theoretical results by providing real-world experiments and training generative adversarial networks on multiple GPUs.

## 1 INTRODUCTION

The surge of deep learning across tasks beyond image classification has triggered a vast literature of optimization paradigms, which transcend the standard empirical risk minimization. For example, training generative adversarial networks (GANs) gives rise to solving a more complicated zero-sum game between a generator and a discriminator (Goodfellow et al., 2020). This can become even more complex when the generator and the discriminator do not have completely antithetical objectives and e.g., constitute a more general game-theoretic setup. A powerful unifying framework which includes those important problems as special cases is *monotone variational inequality (VI)*. Formally, given a monotone operator $A : \mathbb{R}^d \to \mathbb{R}^d$, *i.e.,*

$$\langle A(\mathbf{x}) - A(\mathbf{x}'), \mathbf{x} - \mathbf{x}' \rangle \geq 0 \ \text{ for all } \ \mathbf{x}, \mathbf{x}' \in \mathbb{R}^d,$$

our goal is to find some $\mathbf{x}^* \in \mathbb{R}^d$ such that:

$$\langle A(\mathbf{x}^*), \mathbf{x} - \mathbf{x}^* \rangle \geq 0, \ \text{ for all } \ \mathbf{x} \in \mathbb{R}^d. \tag{VI}$$

Several practical problems can be formulated as a (VI) problem including those with convex-like structures, *e.g., convex minimization*, *saddle-point problems*, and *games* (Facchinei & Pang, 2003; Bauschke & Combettes, 2017; Antonakopoulos et al., 2021) with several applications such as auction theory (Syrgkanis et al., 2015), multi-agent and robust reinforcement learning (Pinto et al., 2017), adversarially robust learning (Schmidt et al., 2018), and GANs.

For various tasks, it is widely known that employing *deep neural networks* (DNNs) along with massive datasets leads to significant improvement in terms of learning (Shalev-Shwartz & Ben-David, 2014). However, DNNs can no longer be trained on a single machine. One common solution is to train on multi-GPU systems (Alistarh et al., 2017). Furthermore, in federated learning (FL), multiple clients, *e.g.,* a few hospitals or several cellphones learn a model collaboratively without sharing local data due to privacy risks (Kairouz et al., 2021).

---

[*]These authors contributed equally to this work.
[†]Department of Informatics, University of Oslo. Work performed at EPFL (LIONS) and Aalborg University.
[‡]Laboratory for Information and Inference Systems (LIONS), EPFL.
Open source code will be released at `https://github.com/LIONS-EPFL/qgenx`

To minimize a single empirical risk, SGD is the most popular algorithm due to its flexibility for parallel implementations and excellent generalization performance (Alistarh et al., 2017; Wilson et al., 2017). Data-parallel SGD has delivered tremendous success in terms of scalability: (Zinkevich et al., 2010; Bekkerman et al., 2011; Recht et al., 2011; Dean et al., 2012; Coates et al., 2013; Chilimbi et al., 2014; Li et al., 2014; Duchi et al., 2015; Xing et al., 2015; Zhang et al., 2015; Alistarh et al., 2017; Faghri et al., 2020; Ramezani-Kebrya et al., 2021; Kairouz et al., 2021). Data-parallel SGD reduces computational costs significantly. However, the communication costs for broadcasting huge stochastic gradients is the main performance bottleneck in large-scale settings (Strom, 2015; Alistarh et al., 2017; Faghri et al., 2020; Ramezani-Kebrya et al., 2021; Kairouz et al., 2021).

Several methods have been proposed to accelerate training for classical empirical risk minimization such as gradient (or model update) compression, gradient sparsification, weight quantization/sparsification, and reducing the frequency of communication though local methods (Dean et al., 2012; Seide et al., 2014; Sa et al., 2015; Gupta et al., 2015; Abadi et al., 2016; Alistarh et al., 2017; Wen et al., 2017; Zhou et al., 2018; Bernstein et al., 2018; Faghri et al., 2020; Ramezani-Kebrya et al., 2021; Kairouz et al., 2021). In particular, *unbiased gradient quantization* is interesting due to both enjoying strong theoretical guarantees along with providing communication efficiency on the fly, *i.e.,* convergence under the same hyperparameteres tuned for uncompressed variants while providing substantial savings in terms of communication costs (Alistarh et al., 2017; Faghri et al., 2020; Ramezani-Kebrya et al., 2021). Unlike full-precision data-parallel SGD, where each processor is required to broadcast its local gradient in full-precision, *i.e.,* transmit and receive huge full-precision vectors at each iteration, unbiased quantization requires each processor to transmit only a few communication bits per iteration for each component of the stochastic gradient.

In this work, we propose communication-efficient variants of a general first-order method that achieves the *optimal rate of convergence* with *improved guarantees on the number of communication bits* for monotone VIs and *show distributed training accelerates convergence*. We employ an adaptive step-size and both adaptive and non-adaptive variants of unbiased quantization schemes tailored to VIs.

There exist three major challenges to tackle this problem: 1) how to modify adaptive variants of unbiased quantization schemes tailored to solve general VIs; 2) can we achieve optimal rate of convergence without knowing noise profile and show benefits of distributed training?; 3) can we validate improvements in terms of scalability without compromising accuracy in large-scale settings? We aim to address those challenges and answer all questions in the affirmative:

## 1.1 SUMMARY OF CONTRIBUTIONS

- We propose *quantized* generalized extra-gradient (Q-GenX) family of algorithms, which employs unbiased compression methods tailored to general VI-solvers. Our framework unifies distributed and communication-efficient variants of stochastic dual averaging, stochastic dual extrapolation, and stochastic optimistic dual averaging.
- Without prior knowledge on the noise profile, we provide an adaptive step-size rule for Q-GenX and achieve a fast rate of $\mathcal{O}(1/T)$ under relative noise, and an order-optimal $\mathcal{O}(1/\sqrt{T})$ in the absolute noise case and show that increasing the number of processors accelerates convergence for general monotone VIs.
- We validate our theoretical results by providing real-world experiments and training generative adversarial networks on multiple GPUs.

## 1.2 RELATED WORK

We overview a summary of related work. Complete related work is provided in Appendix B. Adaptive quantization has been used for speech communication (Cummiskey et al., 1973). In machine learning, adapting quantization levels (Faghri et al., 2020), adapting communication frequency in local SGD (Wang & Joshi, 2019), adapting the number of quantization levels (communication budget) over the course of training (Guo et al., 2020; Agarwal et al., 2021), adapting a gradient sparsification scheme over the course of training (Khirirat et al., 2021), and adapting compression parameters across model layers and training iterations (Markov et al., 2022) have been proposed for *minimizing a single empirical risk*. In this paper, we propose communication-efficient generalized extra-gradient family of algorithms with adaptive quantization and adaptive step-size for a general (VI) problem. In the

VI literature, the benchmark method is EG, proposed by Korpelevich (1976), along with its variants including (Nemirovski, 2004; Nesterov, 2007). Rate interpolation between different noise profiles under an adaptive step-size has been explored by Antonakopoulos et al. (2021). However, their results are limited to *centralized and single-GPU settings*. Beznosikov et al. (2021); Kovalev et al. (2022) have proposed communication-efficient algorithms for VIs with finite-sum structure and variance reduction in centralized settings and (strongly) monotone VIs in decentralized settings, respectively. Unlike (Beznosikov et al., 2021; Kovalev et al., 2022), we achieve fast and order-optimal rates with *adaptive step-size* and *adaptive compression* without requiring variance reduction and strong monotoncity, and improve variance and code-length bounds for unbiased and adaptive compression.

## 2 PROBLEM SETUP

Our objective throughout this paper is to solve VI with $A : \mathbb{R}^d \to \mathbb{R}^d$ being a monotone operator[1].

Moreover, in order to avoid trivialities, we make the following mild assumption:

**Assumption 1** (Existence)**.** The set $\mathcal{X}^* := \{\mathbf{x}^* \in \mathbb{R}^d : \mathbf{x}^* \text{ solves (VI)}\}$ is non-empty.

Let $\mathcal{C}$ denote a non-empty compact test domain. A popular performance measure for the evaluation of a candidate solution for VI is the so-called *restricted gap function* defined as:

$$\text{Gap}_{\mathcal{C}}(\hat{\mathbf{x}}) = \sup_{\mathbf{x} \in \mathcal{C}} \langle A(\mathbf{x}), \hat{\mathbf{x}} - \mathbf{x} \rangle. \tag{Gap}$$

Gap is used to measure $\hat{\mathbf{x}}$'s performance mainly because it characterizes the solutions of VI via its zeros. Mathematically speaking, we have the following proposition:

**Proposition 1** (Nesterov 2009)**.** *Let $\mathcal{C}$ be a non-empty convex subset of $\mathbb{R}^d$. Then, the following holds*

1. *$\text{Gap}_{\mathcal{C}}(\hat{\mathbf{x}}) \geq 0$ for all $\hat{\mathbf{x}} \in \mathcal{C}$*

2. *If $\text{Gap}_{\mathcal{C}}(\hat{\mathbf{x}}) = 0$ and $\mathcal{C}$ contains a neighbourhood of $\hat{\mathbf{x}}$, then $\hat{\mathbf{x}}$ is a solution of VI.*

Proposition 1 is an extension of an earlier characterization shown by Nesterov (2007); we refer the reader to (Antonakopoulos et al., 2019; Nesterov, 2009) and references therein.

From an algorithmic perspective, we primarily consider the generic family of iterative methods, which have access to a *stochastic first-order* oracle, *i.e.,* a black-box feedback mechanism (Nesterov, 2004). The respective iterative algorithm can call the oracle over and over at a (possibly) random sequence of points $\mathbf{x}_0, \mathbf{x}_1, \ldots$ When called at $\mathbf{x}$, the oracle draws an i.i.d. sample $\omega$ from a (complete) probability space $(\Omega, \mathcal{F}, \mathbb{P})$ and returns a *stochastic dual vector* $g(\mathbf{x}; \omega)$ given by

$$g(\mathbf{x}; \omega) = A(\mathbf{x}) + U(\mathbf{x}; \omega) \tag{2.1}$$

where $U(\mathbf{x}; \omega)$ denotes the measurement error or noise. We consider two important noise profile models, i.e., *absolute* and *relative* noise models formally described in the following assumptions:

**Assumption 2** (Absolute noise)**.** Let $\mathbf{x} \in \mathbb{R}^d$ and $\omega \sim \mathbb{P}$. The oracle $g(\mathbf{x}; \omega)$ enjoys these properties: 1) *Almost sure boundedness:* There exists some $M > 0$ such that $\|g(\mathbf{x}; \omega)\|_* \leq M$ a.s.; 2) *Unbiasedness:* $\mathbb{E}\left[g(\mathbf{x}; \omega)\right] = A(\mathbf{x})$; 3) *Bounded absolute variance:* $\mathbb{E}\left[\|U(\mathbf{x}; \omega)\|_*^2\right] \leq \sigma^2$.

The conditions in Assumption 2 are mild and hold for standard oracles, in particular, in the context of adaptive algorithms (Kavis et al., 2019; Levy et al., 2018; Bach & Levy, 2019; Antonakopoulos & Mertikopoulos, 2021) and typically guarantee a convergence rates in $\mathcal{O}(1/\sqrt{T})$ (Nemirovski et al., 2009; Juditsky et al., 2011; Antonakopoulos et al., 2021). Alternatively, one may consider the *relative* noise model following (Polyak, 1987):

**Assumption 3** (Relative noise)**.** The oracle $g(\mathbf{x}; \omega)$ satisfies: 1) *Almost sure boundedness:* There exists some $M > 0$ such that $\|g(\mathbf{x}; \omega)\|_* \leq M$ a.s; 2) *Unbiasedness:* $\mathbb{E}\left[g(\mathbf{x}; \omega)\right] = A(\mathbf{x})$; 3) *Bounded relative variance:* There exists some $c > 0$ such that $\mathbb{E}\left[\|U(\mathbf{x}; \omega)\|_*^2\right] \leq c\|A(\mathbf{x})\|_*^2$.

While Assumption 2 is enough for obtaining the typical $\mathcal{O}(1/\sqrt{T})$ rate in stochastic settings, Assumption 3 may allow us to recover the well-known order-optimal rate of $\mathcal{O}(1/T)$ in deterministic

---

[1]Notations are provided in Appendix A.

---

**Algorithm 1** Q-GenX: Loops are executed in parallel on processors. At certain steps, each processor computes sufficient statistics of a parametric distribution to estimate distribution of dual vectors.

---

**Input:** Local data, parameter vector (local copy) $X_t, Y_t$, learning rate $\{\gamma_t\}$, and set of update steps $\mathcal{U}$

1 **for** $t = 1$ **to** $T$ **do**
2      **if** $t \in \mathcal{U}$ **then**
3          **for** $i = 1$ **to** $K$ **do**
4             Compute sufficient statistics and update quantization levels $\boldsymbol{\ell}_t$
5      **for** $i = 1$ **to** $K$ **do**
6          Compute $V_{i,t}$, encode $c_{i,t} \leftarrow \mathrm{CODE} \circ \mathrm{Q}\big(Q_{\boldsymbol{\ell}_t}(V_{i,t}); \boldsymbol{\ell}_t\big)$, and broadcast $c_{i,t}$
7          Receive $c_{i,t}$ from each processor $i$ and decode $\hat{V}_{i,t} \leftarrow \mathrm{DEQ} \circ \mathrm{CODE}(c_{i,t}; \boldsymbol{\ell}_t)$
8          Aggregate $X_{t+1/2} \leftarrow X_t - \frac{\gamma_t}{K} \sum_{k=1}^{K} \hat{V}_{k,t}$
9          Compute $V_{i,t+1/2}$, encode $d_{i,t} \leftarrow \mathrm{CODE} \circ \mathrm{Q}\big(Q_{\boldsymbol{\ell}_t}(V_{i,t+1/2}); \boldsymbol{\ell}_t\big)$, and broadcast $d_{i,t}$
10          Receive $d_{i,t}$ from each processor $i$ and decode $\hat{V}_{i,t+1/2} \leftarrow \mathrm{DEQ} \circ \mathrm{CODE}(d_{i,t}; \boldsymbol{\ell}_t)$
11          Aggregate $Y_{t+1} \leftarrow Y_t - \frac{1}{K} \sum_{k=1}^{K} \hat{V}_{k,t+1/2}$ and update $X_{t+1} \leftarrow \gamma_{t+1} Y_{t+1}$

---

settings. Intuitively, this improvement is explained by the fact that the noisy error measurements vanish while we approach a solution of VI. In Appendix J, we highlight random coordinate descent and random player updating as popular examples, which motivate Assumption 3.

## 3 QUANTIZED GENERALIZED EXTRA-GRADIENT

### 3.1 SYSTEM MODEL AND PROPOSED ALGORITHM

We now describe the algorithmic framework unifying communication-efficient variants of generalized extra-gradient (EG) family of algorithms. In particular, we consider a synchronous and distributed system with $K$ processors along the lines of *e.g.,* data-parallel SGD (Alistarh et al., 2017; Faghri et al., 2020; Ramezani-Kebrya et al., 2021; Kairouz et al., 2021). These processors can be cellphones and hospitals in FL or GPU resources in a data center. In multi-GPU systems, processors partition a large dataset among themselves such that each processor keeps only a local copy of the current parameter vector and has access to *independent* and *private stochastic dual vectors*. At each iteration, each processor receives stochastic dual vectors from all other processors and aggregates them. To accelerate training, stochastic dual vectors are first compressed by each processor before broadcasting to other peers and then decompressed before each aggregation step. We focus on unbiased compression where, *in expectation*, the output of the decompression of a compressed vector is the same as the original uncompressed vector. We use $Q_{\boldsymbol{\ell}_t}$ to denote a random and adaptive quantization function where the quantization levels $\boldsymbol{\ell}_t$ may change over time. We use $V_{k,t}$ to denote the original (uncompressed) stochastic dual vector computed by process $k$ at time $t$.

Using multiple processors reduces computational costs significantly. However, communication costs to broadcast huge stochastic dual vectors is the main performance bottleneck in practice (Alistarh et al., 2017). In order to reduce communication costs and improve scalability, each processor receives and aggregates the compressed stochastic dual vectors from all peers to obtain the updated parameter vector. Let $\boldsymbol{\ell}_t = (\ell_0, \ell_1^t, \ldots, \ell_s^t, \ell_{s+1})$ denote the sequence of $s$ quantization levels *optimized* at iteration $t$ with $0 = \ell_0 < \ell_1^t < \cdots < \ell_s^t < \ell_{s+1} = 1$. We now define quantization function $Q_{\boldsymbol{\ell}_t}$:

**Definition 1** (Random quantization function). *Let $s \in \mathbb{Z}_+$ denote the number of quantization levels. Let $u \in [0, 1]$ and $\boldsymbol{\ell}_t = (\ell_0, \ell_1^t, \ldots, \ell_s^t, \ell_{s+1})$ denote the sequence of $s$ quantization levels at iteration $t$ with $0 = \ell_0 < \ell_1^t < \cdots < \ell_s^t < \ell_{s+1} = 1$. Let $\tau(u)$ denote the index of a level such that $\ell_{\tau(u)}^t \leq u < \ell_{\tau(u)+1}^t$. Let $\xi_t(u) = (u - \ell_{\tau(u)}^t)/(\ell_{\tau(u)+1}^t - \ell_{\tau(u)}^t)$ be the relative distance of $u$ to level $\tau(u) + 1$. We define the random function $q_{\boldsymbol{\ell}_t}(u) : [0, 1] \to \{\ell_0, \ell_1^t, \ldots, \ell_s^t, \ell_{s+1}\}$ such that $q_{\boldsymbol{\ell}_t}(u) = \ell_{\tau(u)}^t$ with probability $1 - \xi_t(u)$ and $q_{\boldsymbol{\ell}_t}(u) = \ell_{\tau(u)+1}^t$ with probability $\xi_t(u)$. Let $q \in \mathbb{Z}_+$ and $\mathbf{v} \in \mathbb{R}^d$. We define the random quantization of $\mathbf{v}$ as follows:*

$$Q_{\boldsymbol{\ell}_t}(\mathbf{v}) := \|\mathbf{v}\|_q \cdot \mathbf{s} \odot [q_{\boldsymbol{\ell}_t}(u_1), \ldots, q_{\boldsymbol{\ell}_t}(u_d)]^\top$$

*where $\odot$ denotes the element-wise (Hadamard) product.*

Let $\hat{V}_{k,t} = Q_{\ell_t}(V_{k,t})$ and $\hat{V}_{k,t+1/2} = Q_{\ell_t}(V_{k,t+1/2})$ denote the unbiased and quantized stochastic dual vectors for $k \in [K]$ and $t \in [T]$. We propose *quantized* generalized extra-gradient (Q-GenX) family of algorithms with this update rule:

$$X_{t+1/2} = X_t - \frac{\gamma_t}{K} \sum_{k=1}^{K} \hat{V}_{k,t}$$

$$Y_{t+1} = Y_t - \frac{1}{K} \sum_{k=1}^{K} \hat{V}_{k,t+1/2} \tag{Q-GenX}$$

$$X_{t+1} = \gamma_{t+1} Y_{t+1}$$

where $(\hat{V}_{k,0}, \hat{V}_{k,1}, \ldots)$ and $(\hat{V}_{k,1/2}, \hat{V}_{k,3/2}, \ldots)$ are the sequences of stochastic dual vectors computed and quantized by processor $k \in [K]$. Provided that $V_{k,t}$ and $V_{k,t+1/2}$ are stochastic dual vectors for $k \in [K]$, then $\frac{1}{K} \sum_{k=1}^{K} \hat{V}_{k,t}$ and $\frac{1}{K} \sum_{k=1}^{K} \hat{V}_{k,t+1/2}$ remain *unbiased* stochastic dual vectors.

Q-GenX is described in Algorithm 1. In general, the decoded stochastic dual vectors are likely to be different from the original locally computed stochastic dual vectors.

A particularly appealing feature of (Q-GenX) formulation is that it enables us to unify communication-efficient and distributed variants of a wide range of popular first-order methods for solving VIs. In particular, one may observe that under different choices of $\hat{V}_{k,t}$ and $\hat{V}_{k,t+1/2}$, *communication-efficient* variants of stochastic dual averaging (Nesterov, 2009), stochastic dual extrapolation (Nesterov, 2007), and stochastic optimistic dual averaging (Popov, 1980; Rakhlin & Sridharan, 2013; Hsieh et al., 2021; 2022) in *multi-GPU settings* are special cases of Q-GenX:

**Example 3.1. Distributed stochastic dual averaging:** Consider the case where $\hat{V}_{k,t} \equiv 0$ and $\hat{V}_{k,t+1/2} \equiv \hat{g}_{k,t+1/2} = Q_{\ell_t}(A(X_{t+1/2}) + U_{k,t+1/2})$. This setting yields to $X_{t+1/2} = X_t$ and hence $\hat{g}_{k,t+1/2} = \hat{g}_{k,t} = \hat{V}_{k,t+1/2}$. Therefore, Q-GenX reduces to the communication-efficient stochastic dual averaging scheme:

$$Y_{t+1} = Y_t - \frac{1}{K} \sum_{k=1}^{K} \hat{g}_{k,t} \tag{Quantized DA}$$

$$X_{t+1} = \gamma_{t+1} Y_{t+1}$$

**Example 3.2. Distributed stochastic dual extrapolation:** Consider the case where $\hat{V}_{k,t} \equiv \hat{g}_{k,t} = Q_{\ell_t}(A(X_t) + U_{k,t})$ and $\hat{V}_{k,t+1/2} \equiv \hat{g}_{k,t+1/2} = Q_{\ell_t}(A(X_{t+1/2}) + U_{k,t+1/2})$ are noisy oracle queries at $X_t$ and $X_{t+1/2}$, respectively. Then Q-GenX provides the communication-efficient variant of the Nesterov's stochastic dual extrapolation method (Nesterov, 2007):

$$X_{t+1/2} = X_t - \frac{\gamma_t}{K} \sum_{k=1}^{K} \hat{g}_{k,t}$$

$$Y_{t+1} = Y_t - \frac{1}{K} \sum_{k=1}^{K} \hat{g}_{k,t+1/2} \tag{Quantized DE}$$

$$X_{t+1} = \gamma_{t+1} Y_{t+1}$$

**Example 3.3. Distributed stochastic optimistic dual averaging:** Consider the case $\hat{V}_{k,t} \equiv \hat{g}_{k,t-1/2} = Q_{\ell_t}(A(X_{t-1/2}) + U_{k,t-1/2})$ and $\hat{V}_{k,t+1/2} \equiv \hat{g}_{k,t+1/2} = Q_{\ell_t}(A(X_{t+1/2}) + U_{k,t+1/2})$ are the noisy oracle feedback at $X_{t-1/2}$ and $X_{t+1/2}$, respectively. We then obtain the communication-efficient stochastic optimistic dual averaging method:

$$X_{t+1/2} = X_t - \frac{\gamma_t}{K} \sum_{k=1}^{K} \hat{g}_{k,t-1/2}$$

$$Y_{t+1} = Y_t - \frac{1}{K} \sum_{k=1}^{K} \hat{g}_{k,t+1/2} \tag{Quantized OptDA}$$

$$X_{t+1} = \gamma_{t+1} Y_{t+1}$$

This general formulation of Q-GenX allows us to bring these variants under one umbrella and provide theoretical guarantees for all of them in a unified manner.

## 3.2 Encoding

To further reduce communication costs, we can apply information-theoretically coding schemes on top of quantization. Let $q \in \mathbb{Z}_+$. We first note that a vector $\mathbf{v} \in \mathbb{R}^d$ can be *uniquely* represented by a tuple $(\|\mathbf{v}\|_q, \mathbf{s}, \mathbf{u})$ where $\|\mathbf{v}\|_q$ is the $L^q$ norm of $\mathbf{v}$, $\mathbf{s} := [\text{sgn}(v_1), \dots, \text{sgn}(v_d)]^\top$ consists of signs of the coordinates $v_i$'s, and $\mathbf{u} := [u_1, \dots, u_d]^\top$ with $u_i = |v_i|/\|\mathbf{v}\|_q$ are the normalized coordinates. Note that $0 \leq u_i \leq 1$ for all $i \in [d]$. The overall encoding, i.e., composition of coding and quantization, $\text{CODE} \circ Q(\|\mathbf{v}\|_q, \mathbf{s}, \mathbf{q}_{\boldsymbol{\ell}_t}) : \mathbb{R}_+ \times \{\pm 1\}^d \times \{\ell_0, \ell_1^t, \dots, \ell_s^t, \ell_{s+1}\}^d \to \{0, 1\}^*$ in Algorithm 1 uses a standard floating point encoding with $C_b$ bits to represent the positive scalar $\|\mathbf{v}\|_q$, encodes the sign of each coordinate with one bit, and finally applies an *integer* encoding scheme $\Psi : \{\ell_0, \ell_1^t, \dots, \ell_s^t, \ell_{s+1}\} \to \{0, 1\}^*$ to *efficiently* encode each quantized and normalized coordinate $q_{\boldsymbol{\ell}_t}(u_i)$ with the *minimum* expected code-length. The overall decoding $\text{DEQ} \circ \text{CODE} : \{0, 1\}^* \to \mathbb{R}^d$ first reads $C_b$ bits to reconstruct $\|\mathbf{v}\|_q$. Then it applies $\Psi^{-1} : \{0, 1\}^* \to \{\ell_0, \ell_1^t, \dots, \ell_s^t, \ell_{s+1}\}$ to reconstruct normalized coordinates. The encoding/decoding details are provided in Appendix K.

## 3.3 Adaptive quantization

Instead of using a heuristically chosen sequence of quantization levels, adaptive quantization estimates distribution of uncompressed original vectors, i.e., dual vectors, by computing sufficient statistics of a parametric distribution, optimizes quantization levels to minimize the quantization error, and updates those levels adaptively throughout the course of training as the distribution changes. Let $(\Omega_Q, \mathcal{F}_Q, \mathbb{P}_Q)$ denote a complete probability space. Let $\mathbf{q}_{\boldsymbol{\ell}_t} \sim \mathbb{P}_Q$ represent $d$ variables sampled independently for random quantization in Definition 1. Let $\mathbf{v} \in \mathbb{R}^d$ denote a stochastic dual vector to be quantized. Given $\mathbf{v}$, we measure the quantization error by the variance of vector quantization, which is the trace of the covariance matrix:

$$\mathbb{E}_{\mathbf{q}_{\boldsymbol{\ell}_t}}[\|Q_{\boldsymbol{\ell}_t}(\mathbf{v}) - \mathbf{v}\|_2^2] = \|\mathbf{v}\|_q^2 \sum_{i=1}^d \sigma_Q^2(u_i; \boldsymbol{\ell}_t) \tag{3.1}$$

where $u_i = |v_i|/\|\mathbf{v}\|_q$ and $\sigma_Q^2(u; \boldsymbol{\ell}_t) = \mathbb{E}_{\mathbf{q}_{\boldsymbol{\ell}_t}}[(q_{\boldsymbol{\ell}_t}(u) - u)^2] = (\ell_{\tau(u)+1}^t - u)(u - \ell_{\tau(u)}^t)$ is the variance for a normalized coordinate $u$. We optimize $\boldsymbol{\ell}_t$ by minimizing the quantization variance:

$$\min_{\boldsymbol{\ell}_t \in \mathcal{L}} \mathbb{E}_\omega \mathbb{E}_{\mathbf{q}_{\boldsymbol{\ell}_t}} \left[ \|Q_{\boldsymbol{\ell}_t}(g(\mathbf{x}_t; \omega)) - A(\mathbf{x}_t)\|_*^2 \right]$$

where $\mathcal{L} = \{\boldsymbol{\ell} : \ell_j \leq \ell_{j+1}, \forall j, \ell_0 = 0, \ell_{s+1} = 1\}$ denotes the set of feasible solutions.

Since random quanitzation and random samples are statistically independent, we can solve the following equivalent problem:

$$\min_{\boldsymbol{\ell}_t \in \mathcal{L}} \mathbb{E}_\omega \mathbb{E}_{\mathbf{q}_{\boldsymbol{\ell}_t}} \left[ \|Q_{\boldsymbol{\ell}_t}(g(\mathbf{x}_t; \omega)) - g(\mathbf{x}_t; \omega)\|_2^2 \right] \tag{MinVar}$$

To solve MinVar, we first sample $J$ stochastic dual vectors $\{g(\mathbf{x}_t; \omega_1), \dots, g(\mathbf{x}_t; \omega_J)\}^2$. Let $F_j(r)$ denote the marginal cumulative distribution function (CDF) of normalized coordinates conditioned on observing $\|g(\mathbf{x}_t; \omega_j)\|_q$. By the law of total expectation, MinVar can be approximated by:

$$\min_{\boldsymbol{\ell}_t \in \mathcal{L}} \sum_{j=1}^J \|g(\mathbf{x}_t; \omega_j)\|_q^2 \sum_{i=0}^s \int_{\ell_i}^{\ell_{i+1}} \sigma_Q^2(u; \boldsymbol{\ell}_t) \, \mathrm{d}F_j(u) \equiv \min_{\boldsymbol{\ell}_t \in \mathcal{L}} \sum_{i=0}^s \int_{\ell_i}^{\ell_{i+1}} \sigma_Q^2(u; \boldsymbol{\ell}_t) \, \mathrm{d}\tilde{F}(u) \quad \text{(QAda)}$$

where $\tilde{F}(u) = \sum_{j=1}^J \lambda_j F_j(u)$ is the weighted sum of the conditional CDFs with $\lambda_j = \|g(\mathbf{x}_t; \omega_j)\|_q^2 / \sum_{j=1}^J \|g(\mathbf{x}_t; \omega_j)\|_q^2$.

Finally, we solve QAda efficiently by either updating levels one at a time or gradient descent along the lines of (Faghri et al., 2020).

---

[2] A more fine-grained analysis can be done by considering two sequences of adaptive levels one for $V_{k,t}$'s and another one for $V_{k,t+1/2}$'s. While in this paper, we quantize both sequences with the same quantization scheme, it is possible to further reduce quantization errors by considering two fine-grained quantization schemes at the cost of additional computations at processors.

## 4 THEORETICAL GUARANTEES

We first establish a variance error bound for a general unbiased and normalized compression scheme and a bound on the expected number of communication bits to encode $Q_{\boldsymbol{\ell}}(\mathbf{v})$, *i.e.*, the output of $\text{CODE} \circ \text{Q}$, which is introduced in Section 3.2. The detailed proofs are provided in the appendix.

Let $\bar{\ell} := \max_{1 \leq j \leq s} \ell_{j+1}/\ell_j$ and $d_{\text{th}} = (2/\ell_1)^{\min\{q,2\}}$. Under general $L^q$ normalization and sequence of $s$ quantization levels $\boldsymbol{\ell}$, we first establish an upper bound on the variance of quantization:

**Theorem 1** (Variance bound). *Let $\mathbf{v} \in \mathbb{R}^d$, $q \in \mathbb{Z}_+$, and $s \in \mathbb{Z}_+$. Let $\boldsymbol{\ell} = (\ell_0, \ldots, \ell_{s+1})$ denote a sequence of $s$ quantization levels defined in Definition 1. The quantization of $\mathbf{v}$ in Definition 1 is unbiased, i.e., $\mathbb{E}_{\mathbf{q}_{\boldsymbol{\ell}}}[Q_{\boldsymbol{\ell}}(\mathbf{v})] = \mathbf{v}$. Furthermore, we have*

$$\mathbb{E}_{\mathbf{q}_{\boldsymbol{\ell}}}[\|Q_{\boldsymbol{\ell}}(\mathbf{v}) - \mathbf{v}\|_2^2] \leq \epsilon_Q \|\mathbf{v}\|_2^2, \tag{4.1}$$

*where $\epsilon_Q = \frac{\bar{\ell} + \bar{\ell}^{-1}}{4} + \frac{1}{4}\ell_1^2 d^{\frac{2}{\min\{q,2\}}} \mathbb{1}\{d \leq d_{\text{th}}\} + \left(\ell_1 d^{\frac{1}{\min\{q,2\}}} - 1\right) \mathbb{1}\{d \geq d_{\text{th}}\} - \frac{1}{2}$ and $\mathbb{1}$ is the indicator function.*

Theorem 1 implies that if $g(\mathbf{x}; \omega)$ is an unbiased stochastic dual vector with a bounded absolute variance $\sigma^2$, then $Q_{\boldsymbol{\ell}}(g(\mathbf{x}; \omega))$ will be an unbiased stochastic dual vector with a variance upper bound $\epsilon_Q \sigma^2$. Note that, the dominant term $\frac{1}{4}\ell_1^2 d^{\frac{2}{\min\{q,2\}}} \mathbb{1}\{d \leq d_{\text{th}}\} + \left(\ell_1 d^{\frac{1}{\min\{q,2\}}} - 1\right) \mathbb{1}\{d \geq d_{\text{th}}\}$ monotonically decreases as the number of quantization levels increases. Unlike (Alistarh et al., 2017, Theorem 3.2) and (Ramezani-Kebrya et al., 2021, Theorem 4) that hold under the special cases of $L^2$ normalization with uniform and exponentially spaced levels, respectively, our bound in Theorem 1 holds under general $L^q$ normalization and arbitrary sequence of quantization levels. For the special case of $L^2$ normalization in the regime of large $d$, which is the case in practice, our bound in Theorem 1 is $\mathcal{O}(\ell_1 \sqrt{d})$, which is arbitrarily smaller than $\mathcal{O}(\sqrt{d}/s)$ and $\mathcal{O}(2^{-s}\sqrt{d})$ in (Alistarh et al., 2017, Theorem 3.2) and (Ramezani-Kebrya et al., 2021, Theorem 4), respectively, because $\ell_1$ is adaptively designed to minimize the variance of quantization. Furthermore, unlike the bound in (Faghri et al., 2020, Theorem 2) that requires an inner optimization problem over an auxiliary parameter $p$, the bound in Theorem 1 is provided in an explicit form without an inner optimization problem, which matches the known $\Omega(\sqrt{d})$ lower bound.

**Theorem 2** (Code-length bound). *Let $p_j$ denote the probability of occurrence of $\ell_j$ (weight of symbol $\ell_j$) for $j \in [s]$. Under the setting specified in Theorem 1, the expectation $\mathbb{E}_\omega \mathbb{E}_{\mathbf{q}_{\boldsymbol{\ell}}}[|\text{CODE} \circ \text{Q}(Q_{\boldsymbol{\ell}}(g(\mathbf{x}; \omega)); \boldsymbol{\ell})|]$ of the number of bits to encode $Q_{\boldsymbol{\ell}}(g(\mathbf{x}; \omega))$ is bounded by*

$$\mathbb{E}_\omega \mathbb{E}_{\mathbf{q}_{\boldsymbol{\ell}}}[|\text{CODE} \circ \text{Q}(Q_{\boldsymbol{\ell}}(g(\mathbf{x}; \omega)); \boldsymbol{\ell})|] = \mathcal{O}\left((\sum_{j=1}^s p_j \log(1/p_j) - p_0)d\right). \tag{4.2}$$

Note $\{p_0, \ldots, p_{s+1}\}$ can be computed efficiently using the weighted sum of the conditional CDFs in QAda and quantization levels. We provide their expressions in Appendix E. Unlike (Alistarh et al., 2017, Theorem 3.4) and (Ramezani-Kebrya et al., 2021, Theorem 5) that hold under the special cases of $L^2$ normalization, our bound in Theorem 2 holds under general $L^q$ normalization. For the special case of $L^2$ normalization with $s = \sqrt{d}$ as in (Alistarh et al., 2017, Theorem 3.4), our bound in Theorem 2 can be arbitrarily smaller than (Alistarh et al., 2017, Theorem 3.4) and (Ramezani-Kebrya et al., 2021, Theorem 5) depending on $\{p_0, \ldots, p_{s+1}\}$. Compared to (Faghri et al., 2020, Theorem 3), our bound in Theorem 2 does not have an additional $n_{\ell_1,d}$ term. We show that a total expected number of $\mathcal{O}(Kd/\epsilon)$ bits are required to reach an $\epsilon$ gap, which matches the lower bound developed for convex optimization problems with finite-sum structures (Tsitsiklis & Luo, 1987; Korhonen & Alistarh, 2021).

We finally present the convergence guarantees for Q-GenX given access to stochastic dual vectors under both *absolute noise* and *relative noise* models in Assumptions 2 and 3, respectively.

**Theorem 3** (Q-GenX under absolute noise). *Let $\mathcal{C} \subset \mathbb{R}^d$ denote a compact neighborhood of a solution for (VI) and let $D^2 := \sup_{X \in \mathcal{C}} \|X - X_0\|^2$. Suppose that the oracle and the problem (VI) satisfy Assumptions 1 and 2, respectively, Algorithm 1 is executed for $T$ iterations on $K$ processors with an adaptive step-size $\gamma_t = K(1 + \sum_{i=1}^{t-1} \sum_{k=1}^K \|\hat{V}_{k,i} - \hat{V}_{k,i+1/2}\|^2)^{-1/2}$, and quantization levels are updated $J$ times where $\ell_j$ with variance bound $\epsilon_{Q,j}$ in (4.1) and code-length bound $N_{Q,j}$*

*in* (4.2) *is used for $T_j$ iterations with $\sum_{j=1}^{J} T_j = T$. Then we have*

$$\mathbb{E}\Big[\operatorname{Gap}_{\mathcal{C}}\Big(\frac{1}{T}\sum_{t=1}^{T} X_{t+1/2}\Big)\Big] = \mathcal{O}\Big(\frac{(\sum_{j=1}^{J}\sqrt{\epsilon_{Q,j}T_j/T}M + \sigma)D^2}{\sqrt{TK}}\Big).$$

*In addition, Algorithm 1 requires each processor to send at most $\frac{2}{T}\sum_{j=1}^{J} T_j N_{Q,j}$ communication bits per iteration in expectation.*

We now establish *fast rate* of $\mathcal{O}(1/T)$ under *relative noise* and a mild regularity condition:

**Assumption 4** (Co-coercivity). Let $\beta > 0$. We assume that operator $A$ is $\beta$-cocoercive:

$$\langle A(\mathbf{x}) - A(\mathbf{x}'), \mathbf{x} - \mathbf{x}'\rangle \geq \beta\|A(\mathbf{x}) - A(\mathbf{x}')\|_*^2 \quad \text{for all} \quad \mathbf{x}, \mathbf{x}' \in \mathbb{R}^d. \tag{4.3}$$

For a panoramic view of this class of operators, we refer the reader to (Bauschke & Combettes, 2017).

*Remark* 1. The order-optimal rate of $\mathcal{O}(1/\sqrt{T})$ under absolute noise does not require co-cocercivity. Our adaptive step-size also does not depend on the noise model or co-coercivity. Co-coercivity is required only to achieve *fast rate* of $\mathcal{O}(1/T)$ in the case of *relative noise*.

**Theorem 4** (Q-GenX under relative noise). *Let $\mathcal{C} \subset \mathbb{R}^d$ denote a compact neighborhood of a solution for* (VI) *and let $D^2 := \sup_{X \in \mathcal{C}} \|X - X_0\|^2$. Suppose that the oracle and the problem* (VI) *satisfy Assumptions 1, 3, and 4, Algorithm 1 is executed for $T$ iterations on $K$ processors with an adaptive step-size $\gamma_t = K(1 + \sum_{i=1}^{t-1}\sum_{k=1}^{K}\|\hat{V}_{k,i} - \hat{V}_{k,i+1/2}\|^2)^{-1/2}$, and quantization levels are updated $J$ times where $\ell_j$ with variance bound $\epsilon_{Q,j}$ in* (4.1) *and code-length bound $N_{Q,j}$ in* (4.2) *is used for $T_j$ iterations with $\sum_{j=1}^{J} T_j = T$. Then we have*

$$\mathbb{E}\Big[\operatorname{Gap}_{\mathcal{C}}\Big(\frac{1}{T}\sum_{t=1}^{T} X_{t+1/2}\Big)\Big] = \mathcal{O}\Big(\frac{((c+1)\sum_{j=1}^{J} T_j\epsilon_{Q,j}/T + c)D^2}{KT}\Big).$$

*In addition, Algorithm 1 requires each processor to send at most $\frac{2}{T}\sum_{j=1}^{J} T_j N_{Q,j}$ communication bits per iteration in expectation.*

To the best of our knowledge, our results in Theorems 3 and 4 are the first ones proving that increasing the number of processors accelerates convergence for general monotone VIs under an adaptive step size. Theorems 3 and 4 show that we can attain a fast rate of $\mathcal{O}(1/T)$ and an order-optimal $\mathcal{O}(1/\sqrt{T})$ without prior knowledge on the noise profile while significantly reducing communication costs.

Compared to saddle point problems, our rates are optimal. This can be verified by lower bounds in (Beznosikov et al., 2020). For convex problems in deterministic settings, the rate can be improved to $\mathcal{O}(1/T^2)$ via acceleration. However, it is known that in the stochastic and distributed settings, our rates cannot be improved even with acceleration. E.g., for convex and smooth problems under absolute noise model, the lower bound of $\Omega(\frac{1}{\sqrt{TK}})$ can be established by (Woodworth et al., 2021, Theorem 1) and setting the number of gradients per round to one.

In Appendix I, we build on Theorems 3 and 4 to capture the trade-off between the number of iterations to converge and time per iteration, which includes total time required to update a model on each GPU.

## 5 EXPERIMENTAL EVALUATION

In order to validate our theoretical results, we build on the code base of Gidel et al. (2019) and run an instantiation of Q-GenX obtained by combining ExtraAdam with the compression offered by the `torch_cgx` pytorch extension of Markov et al. (2022), and train a WGAN-GP (Arjovsky et al., 2017) on CIFAR10 (Krizhevsky, 2009).

Since `torch_cgx` uses OpenMPI (Gabriel et al., 2004) as its communication backend, we use OpenMPI as the communication backend for the full gradient as well for a fairer comparison. We deliberately do *not* tune any hyperparameters to fit the larger batchsize since simliar to (Gidel et al., 2019), we do not claim to set a new SOTA with these experiments but simply want to show that our theory holds up in practice and can potentially lead to improvements. For this, we present a basic

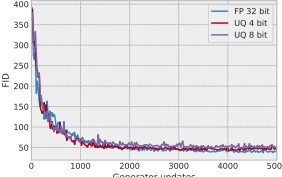 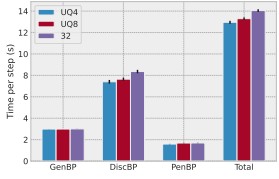

| Mode | GenBP | DiscBP | PenBP | Total |
|------|-------|--------|-------|-------|
| UQ4 | 2.99 | 7.40 | 1.59 | 12.96 |
| UQ8 | 2.99 | 7.65 | 1.69 | 13.29 |
| FP32 | 3.00 | 8.36 | 1.69 | 14.05 |

**Figure 1:** FID evolution during training (left). We compare full-precision ExtraAdam with a simple instantiation of Q-GenX. FID stands for Frechet inception distance, which is a standard GAN quality metric introduced in (Heusel et al., 2017). Fine grained comparison of average `.backward()` times on generator, discriminator, gradient penalty as well as total training time (s) (middle and right). The `.backward()` function is where pytorch DistributedDataParallel (DDP) handles gradient exchange.

experiment showing that even for a *very small problem size* and *a heuristic base compression method of cgx*, we can achieve a *noticeable speedup* of around $8\%$. We expect further gains to be achievable for larger problems and more advanced compression methods. Given differences in terms of settings and the *lack of any code, let alone an efficient implementations that can be used in a real-world setting (i.e. CUDA kernels integrated with networking)*, it is difficult to impossible to conduct a fair comparison with Beznosikov et al. (2021). More details and a comparison with QSGDA of Beznosikov et al. (2022) are provided in Appendix H. We are not aware of any other existing method dealing with the same problem in our paper without the cost associated with variance reduction.

We follow exactly the setup of (Gidel et al., 2019) except that we share an effective batch size of 1024 across 3 nodes (strong scaling) connected via Ethernet, and use Layernorm (Ba et al., 2016) instead of Batchnorm (Ioffe & Szegedy, 2015) since Batchnorm is known to be challenging to work with in distributed training as well as interacting badly with the WGAN-GP penalty. The results are shown in Figure 1 (left) showing evolution of FID. We note that we do *not* scale the learning rate or any other hyperparameters to account for these two changes so this experiment is *not* meant to claim SOTA performance, merely to illustrate that

1. Even with the simplest possible unbiased quantization on a relatively small-scale setup, we can observe a speedup (about $8\%$).

2. This speedup does not drastically change the performance.

We compare training using the full gradient of 32 bit (FP32) to training with gradients compressed to 8 (UQ8) and 4 bits (UQ4) using a bucket size of 1024. Figure 1 (middle and right) shows a more fine grained breakdown of the time used for back propagation (BP) where the network activity takes place. GenBP, DiscBP and PenBP refer to the backpropagation for generator, discriminator, and the calculation of the gradient penalty, respectively. Total refers to the sum of these times.

## 6 CONCLUSIONS

We have considered mononote VIs in a synchronous and multi-GPU setting where multiple processors compute independent and private stochastic dual vectors in parallel. We proposed Q-GenX, which employs unbiased and adaptive compression methods tailored to a generic unifying framework for solving VIs. Without knowing the noise profile in advance, we have obtained an adaptive step-size rule, which achieves a fast rate of $\mathcal{O}(1/T)$ under relative noise, and an order-optimal $\mathcal{O}(1/\sqrt{T})$ in the absolute noise case along with improved guarantees on the expected number of communication bits. Our results show that increasing the number of processors accelerates convergence.

Developing new VI-solvers for *asynchronous* settings and establishing convergence guarantees while relaxing co-coercivity assumption on the operator are interesting problems left for future work. Expected co-coercivity has been used as a more relaxed noise model (Loizou et al., 2021). It is interesting to study monotone VIs under expected co-coercivity and adaptive step-sizes in the future.

ACKNOWLEDGMENTS

The authors would like to thank Fartash Faghri, Yang Linyan, Ilia Markov, and Hamidreza Ramezanikebrya for helpful discussions. This project has received funding from the European Research Council (ERC) under the European Union's Horizon 2020 research and innovation programme (grant agreement n° 725594 - time-data). This work was supported by the Swiss National Science Foundation (SNSF) under grant number 200021_205011.

The work of Ali Ramezani-Kebrya was in part supported by the Research Council of Norway, through its Centre for Research-based Innovation funding scheme (Visual Intelligence under grant no. 309439), and Consortium Partners.

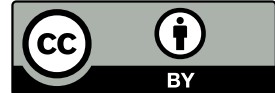

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

## A  APPENDIX

**Notation.** We use $\mathbb{E}[\cdot]$, $\|\cdot\|$, $\|\cdot\|_0$, and $\|\cdot\|_*$ to denote the expectation operator, Euclidean norm, number of nonzero elements of a vector, and dual norm, respectively. We use $|\cdot|$ to denote the length of a binary string, the length of a vector, and cardinality of a set. We use lower-case bold letters to denote vectors. Sets are typeset in a calligraphic font. The base-2 logarithm is denoted by $\log$, and the set of binary strings is denoted by $\{0, 1\}^*$. We use $[n]$ to denote $\{1, \ldots, n\}$ for an integer $n$.

**Content of the appendix.** The appendix is organized as follows:

- Complete related work is discussed in Appendix B.
- Special cases of Q-GenX are provided in Appendix C.
- Theorem 1 (variance bound) is proved in Appendix D.
- Theorem 2 (code-length bound) is proved in Appendix E.
- Theorem 3 (Q-GenX under absolute noise) is proved in Appendix F.
- Theorem 4 (Q-GenX under relative noise) is proved in Appendix G.
- Additional experimental details are included in Appendix H.
- Trade-off between number of iterations and time per iteration is provided in  Appendix I.
- Popular Examples motivating Assumption 3 are provided in Appendix J.
- The encoding/decoding details are provided in Appendix K.

## B   FURTHER RELATED WORK

**Unbiased compression.** Seide et al. (2014) proposed SignSGD, an efficient heuristic scheme to reduce communication costs drastically by quantizing each gradient component to two values. (This scheme is sometimes termed as 1bitSGD (Seide et al., 2014).) Bernstein et al. (2018) later provided convergence guarantees for a variant of SignSGD. Note that the quantization employed by SignSGD is not unbiased, and so a new analysis was required. Alistarh et al. (2017) proposed quantized SGD (QSGD) focusing on the uniform quantization of stochastic gradients normalized to have unit Euclidean norm. Their experiments illustrate a similar quantization method, where gradients are normalized to have unit $L^\infty$ norm, achieves better performance. We refer to this method as QSGDinf or Qinf in short. Wen et al. (2017) proposed TernGrad, which can be viewed as a special case of QSGDinf with three quantization levels. Ramezani-Kebrya et al. (2021) proposed nonuniform quantization levels (NUQSGD) and demonstrated superior empirical results compared to QSGDinf. Recently, lattice-based quantization has been studied for distributed mean estimation and variance reduction (Davies et al., 2021).

Adaptive quantization has been used for speech communication and storage (Cummiskey et al., 1973). In machine learning, several biased and unbiased schemes have been proposed to compress networks and gradients. In this work, we focus on unbiased and coordinate-wise schemes to compress gradients. Zhang et al. (2017) proposed ZipML, which is an optimal quantization method if all points to be quantized are known a priori. To find the optimal sequence of quantization levels, a dynamic program is solved whose computational and memory cost is quadratic in the number of points to be quantized, which in the case of gradients would correspond to their dimension. Faghri et al. (2020) have proposed two adaptive gradient compression schemes where multiple processors update their compression schemes in parallel by efficiently computing sufficient statistics of a parametric distribution. Adaptive quantization methods in (Faghri et al., 2020) are applicable only when minimizing a single empirical risk. In addition, convergence guarantees in (Faghri et al., 2020) are established for smooth nonconvex optimization with a fixed step-size. In this paper, we propose communication-efficient variants of generalized EG family of algorithms with an adaptive and unbiased quantization scheme for a general (VI) problem. Furthermore, we establish improved variance and code-length bounds and optimal convergence guarantees for nonconvex problems with an adaptive step-size.

Adaptive gradient compression has been studied in other contexts when minimizing a single empirical risk, such as adapting the communication frequency in local SGD (Wang & Joshi, 2019), adapting the number of quantization levels (communication budget) over the course of training (Guo et al., 2020; Agarwal et al., 2021), adapting a gradient sparsification scheme over the course of training (Khirirat et al., 2021), and adapting compression parameters across model layers and training iterations (Markov et al., 2022). We focus on unbiased and normalized quantization and adapt quantization levels to *minimize quantization error* for generalized EG family of algorithms, which has not been considered in the literature.

**First-order methods to solve VIs.** In the VI literature, the benchmark method is extragradient (EG), proposed by Korpelevich (1976), along with its variants including (Nemirovski, 2004; Nesterov, 2007). Furthermore, there have been a line of work that aims to focus on establishing convergence guarantees though adopting an adaptive step-size policy. To that end, we review the most relevant works below.

For unconstrained problems with an operator that is locally Lipschitz continuous (but not necessarily globally), the Golden Ratio Algorithm (GRAAL) of Malitsky (2020) achieves convergence without requiring prior knowledge of the problem's Lipschitz parameter. Moreover, such guarantees are provided in problems with a bounded domain by the Generalized Mirror Prox (GMP) algorithm of Stonyakin et al. (2018) under the umbrella of Hölder continuity.

A more relevant method that simultaneously achieves an $\mathcal{O}(1/\sqrt{T})$ rate in non-smooth and/or stochastic problems and an $\mathcal{O}(1/T)$ rate in smooth ones is the recent algorithm of Bach & Levy (2019). This algorithm employs an adaptive, AdaGrad-like step-size policy which allows the method to interpolate between these regimes. On the negative side, this algorithm requires a bounded domain with a (Bregman) diameter that is known in advance.

In optimization community, similar rate interpolation guarantees between different noise profiles has been explored by Antonakopoulos et al. (2021). However, their results are limited to *centralized, single GPU settings*.

While all these algorithms concern standard single-GPU settings with having access to full-precision stochastic dual vectors, we consider multi-GPU settings, which has not been considered before.

Beznosikov et al. (2021); Kovalev et al. (2022) have proposed communication-efficient algorithms for VIs with finite-sum structure and variance reduction in centralized settings and (strongly) monotone VIs in decentralized settings, respectively. Unlike (Beznosikov et al., 2021; Kovalev et al., 2022), we achieve fast and order-optimal rates with *adaptive step-size* and *adaptive compression* without requiring variance reduction and strong monotoncity, and improve variance and code-length bounds for unbiased and adaptive compression.

**Detailed comparison with (Antonakopoulos et al., 2021; Alistarh et al., 2017; Faghri et al., 2020; Ramezani-Kebrya et al., 2021).** In this section, we elaborate and provide a detailed comparison with the most relevant related work (Antonakopoulos et al., 2021; Alistarh et al., 2017; Faghri et al., 2020; Ramezani-Kebrya et al., 2021).

Although our results build upon the extra-gradient literature Korpelevich (1976), as Antonakopoulos et al. (2021) does, in this paper, we address *monotone VI/ convex-concave min-max problems* in *distributed and large-scale settings*, which has not been considered in Antonakopoulos et al. (2021) that strictly refers to a strictly single-GPU and centralized setting. The considered distributed framework complicates the analysis in a significant manner, since we have to simultaneously treat two different *types* of randomness. In particular, on one hand, we face randomness associated with the compression scheme (which is necessary to achieve substantial communication savings in a distributed setup) where on the other hand we deal with different noisy feedback models stemming from inexact operator calculations (before any compression takes place), which together result in efficient implementations at each GPU. We show benefits of distributed training in terms of accelerating convergence for general monotone VIs.

Unlike (Alistarh et al., 2017, Theorem 3.2) and (Ramezani-Kebrya et al., 2021, Theorem 4) that hold under the special cases of $L^2$ normalization with uniform and exponentially spaced levels, respectively, our bound in Theorem 1 holds under general $L^q$ normalization and arbitrary sequence of quantization levels. For the special case of $L^2$ normalization in the regime of large $d$, which is the case in practice, our bound in Theorem 1 is $\mathcal{O}(\ell_1\sqrt{d})$, which is arbitrarily smaller than $\mathcal{O}(\sqrt{d}/s)$ and $\mathcal{O}(2^{-s}\sqrt{d})$ in (Alistarh et al., 2017, Theorem 3.2) and (Ramezani-Kebrya et al., 2021, Theorem 4), respectively, because $\ell_1$ is adaptively designed to minimize the variance of quantization. Furthermore, unlike the bound in (Faghri et al., 2020, Theorem 2) that requires an inner optimization problem over an auxiliary parameter $p$, the bound in Theorem 1 is provided in an explicit form without an inner optimization problem, which matches the lower bound.

Unlike (Alistarh et al., 2017, Theorem 3.4) and (Ramezani-Kebrya et al., 2021, Theorem 5) that hold under the special cases of $L^2$ normalization, our code-length bound in Theorem 2 holds under general $L^q$ normalization. For the special case of $L^2$ normalization with $s = \sqrt{d}$ as in (Alistarh et al., 2017, Theorem 3.4), our bound in Theorem 2 can be arbitrarily smaller than (Alistarh et al., 2017, Theorem 3.4) and (Ramezani-Kebrya et al., 2021, Theorem 5) depending on $\{p_0, \ldots, p_{s+1}\}$. Compared to (Faghri et al., 2020, Theorem 3), our bound in Theorem 2 does not have an additional $n_{\ell_1,d}$ term.

## C   Special cases of Q-GenX

In this section, we show that under different choices of $V_{k,t}$ and $V_{k,t+1/2}$, one can obtain *communication-efficient* variants of stochastic dual averaging (Nesterov, 2009), stochastic dual extrapolation (Nesterov, 2007), and stochastic optimistic dual averaging (Popov, 1980; Rakhlin & Sridharan, 2013; Hsieh et al., 2021; 2022) in *multi-GPU settings* as special cases of Q-GenX.

**Example C.1. Communication-efficient stochastic dual averaging:** Consider the case where $\hat{V}_{k,t} \equiv 0$ and $\hat{V}_{k,t+1/2} \equiv \hat{g}_{k,t+1/2} = Q_{\boldsymbol{\ell}_t}(A(X_{t+1/2}) + U_{k,t+1/2})$. This setting yields to $X_{t+1/2} = X_t$ and hence $\hat{g}_{k,t+1/2} = \hat{g}_{k,t} = \hat{V}_{k,t+1/2}$. Therefore, Q-GenX reduces to the communication-

efficient stochastic dual averaging scheme:

$$Y_{t+1} = Y_t - K^{-1} \sum_{k=1}^{K} \hat{g}_{k,t} \qquad \text{(Quantized DA)}$$

$$X_{t+1} = \gamma_{t+1} Y_{t+1}$$

**Example C.2. Communication-efficient stochastic dual extrapolation:** Consider the case where $\hat{V}_{k,t} \equiv \hat{g}_{k,t} = Q_{\boldsymbol{\ell}_t}(A(X_t) + U_{k,t})$ and $\hat{V}_{k,t+1/2} \equiv \hat{g}_{k,t+1/2} = Q_{\boldsymbol{\ell}_t}(A(X_{t+1/2}) + U_{k,t+1/2})$ are noisy oracle queries at $X_t$ and $X_{t+1/2}$, respectively. Then Q-GenX provides the communication-efficient variant of Nesterov's stochastic dual extrapolation method (Nesterov, 2007):

$$X_{t+1/2} = X_t - \frac{\gamma_t}{K} \sum_{k=1}^{K} \hat{g}_{k,t}$$

$$Y_{t+1} = Y_t - K^{-1} \sum_{k=1}^{K} \hat{g}_{k,t+1/2} \qquad \text{(Quantized DE)}$$

$$X_{t+1} = \gamma_{t+1} Y_{t+1}$$

**Example C.3. Communication-efficient stochastic optimistic dual averaging:** Consider the case $\hat{V}_{k,t} \equiv \hat{g}_{k,t-1/2} = Q_{\boldsymbol{\ell}_t}(A(X_{t-1/2}) + U_{k,t-1/2})$ and $\hat{V}_{k,t+1/2} \equiv \hat{g}_{k,t+1/2} = Q_{\boldsymbol{\ell}_t}(A(X_{t+1/2}) + U_{k,t+1/2})$ are the noisy oracle feedback at $X_{t-1/2}$ and $X_{t+1/2}$, respectively. We then obtain the communication-efficient stochastic optimistic dual averaging method:

$$X_{t+1/2} = X_t - \frac{\gamma_t}{K} \sum_{k=1}^{K} \hat{g}_{k,t-1/2}$$

$$Y_{t+1} = Y_t - K^{-1} \sum_{k=1}^{K} \hat{g}_{k,t+1/2} \qquad \text{(Quantized OptDA)}$$

$$X_{t+1} = \gamma_{t+1} Y_{t+1}$$

## D    PROOF OF THEOREM 1 (VARIANCE BOUND)

Let $u_j = |v_j|/\|\mathbf{v}\|_q$, $\mathcal{B}_0 := [0, \ell_1]$, and $\mathcal{B}_j := [\ell_j, \ell_{j+1}]$ for $j \in [s]$. Let $\mathsf{V}_{\boldsymbol{\ell}}(\mathbf{v}) = \mathbb{E}_{\mathbf{q}_{\boldsymbol{\ell}}}[\|Q_{\boldsymbol{\ell}}(\mathbf{v}) - \mathbf{v}\|_2^2]$ denote the variance of quantization in Eq. (3.1). Then we have

$$\mathsf{V}_{\boldsymbol{\ell}}(\mathbf{v}) = \|\mathbf{v}\|_q^2 \Big( \sum_{u_i \in \mathcal{B}_0} (\ell_1 - u_i) u_i + \sum_{j=1}^{s} \sum_{u_i \in \mathcal{B}_j} (\ell_{j+1} - u_i)(u_i - \ell_j) \Big). \qquad \text{(D.1)}$$

We first find the minimum $k_j$ that satisfies $(\ell_{j+1} - u)(u - \ell_j) \le k_j u^2$ for $u \in \mathcal{B}_j$ and $j \in [s]$. The minimum $k_j$ can be obtained by changing the variable $u = \ell_j \theta$:

$$
\begin{aligned}
k_j &= \max_{1 \le \theta \le \ell_{j+1}/\ell_j} \frac{(\ell_{j+1}/\ell_j - \theta)(\theta - 1)}{\theta^2} \\
&= \frac{(\ell_{j+1}/\ell_j - 1)^2}{4(\ell_{j+1}/\ell_j)}.
\end{aligned}
\qquad \text{(D.2)}
$$

We note that $\ell_{j+1}/\ell_j > 1$ and $(x-1)^2/(4x)$ is monotonically increasing function of $x$ for $x > 1$.

Furthermore, note that

$$\sum_{u_i \notin \mathcal{B}_0} u_i^2 \le \frac{\|\mathbf{v}\|_2^2}{\|\mathbf{v}\|_q^2}.$$

Substituting Eq. (D.2) into Eq. (D.1), an upper bound on $\mathtt{V}_{\boldsymbol{\ell}}(\mathbf{v})$ is given by

$$\mathtt{V}_{\boldsymbol{\ell}}(\mathbf{v}) \leq \|\mathbf{v}\|_q^2 \Big( \Big(\frac{\overline{\ell} + \overline{\ell}^{-1}}{4} - \frac{1}{2}\Big)\frac{\|\mathbf{v}\|_2^2}{\|\mathbf{v}\|_q^2} + \sum_{u_i \in \mathcal{B}_0}(\ell_1 - u_i)u_i\Big).$$

In the rest of the proof, we use the following known lemma.

**Lemma 1.** *Let $\mathbf{v} \in \mathbb{R}^d$. Then, for all $0 < p < q$, we have $\|\mathbf{v}\|_q \leq \|\mathbf{v}\|_p \leq d^{1/p-1/q}\|\mathbf{v}\|_q$.*

We note that Lemma 1 holds even when $q < 1$ and $\|\cdot\|_q$ is merely a seminorm.

We now establish an upper bound on $\sum_{u_i \in \mathcal{B}_0}(2^{-s} - u_i)u_i$.

**Lemma 2** (Ramezani-Kebrya et al. 2021, Lemma 15). *Let $p \in (0,1)$ and $u \in \mathcal{B}_0$. Then we have $u(\ell_1 - u) \leq K_p \ell_1^{(2-p)}u^p$ where*

$$K_p = \Big(\frac{1/p}{2/p - 1}\Big)\Big(\frac{1/p - 1}{2/p - 1}\Big)^{(1-p)}. \tag{D.3}$$

Let $\mathcal{S}_j$ denote the coordinates of vector $\mathbf{v}$ whose elements fall into the $(j+1)$-th bin, *i.e.*, $\mathcal{S}_j := \{i : u_i \in \mathcal{B}_j\}$ for $j \in [s]$. For any $0 < p < 1$ and $q \geq 2$, we have

$$\begin{aligned}
\|\mathbf{v}\|_q^2 \sum_{u_i \in \mathcal{B}_0} u_i^p &= \|\mathbf{v}\|_q^{2-p} \sum_{i \in \mathcal{S}_0} |v_i|^p \\
&\leq \|\mathbf{v}\|_q^{2-p}\|\mathbf{v}\|_p^p \\
&\leq \|\mathbf{v}\|_q^{2-p}\|\mathbf{v}\|_2^p d^{1-p/2} \\
&\leq \|\mathbf{v}\|_2^2 d^{1-p/2},
\end{aligned}$$

where the third inequality holds as $\|\mathbf{v}\|_p \leq \|\mathbf{v}\|_2 d^{1/p-1/2}$ using Lemma 1 and the last inequality holds as $\|\mathbf{v}\|_q \leq \|\mathbf{v}\|_2$ for $q \geq 2$. Using Lemmas 1 and 2, we establish an upper bound on $\mathtt{V}_{\boldsymbol{\ell}}(\mathbf{v})$:

$$\mathtt{V}_{\boldsymbol{\ell}}(\mathbf{v}) \leq \|\mathbf{v}\|_2^2\Big(\frac{\overline{\ell} + \overline{\ell}^{-1}}{4} - \frac{1}{2} + K_p\ell_1^{(2-p)}d^{1-p/2}\Big).$$

For $q \geq 1$, we note that $\|\mathbf{v}\|_q^{2-p} \leq \|\mathbf{v}\|_2^{2-p}d^{\frac{2-p}{\min\{q,2\}} - \frac{2-p}{2}}$, *i.e.*,

$$\mathtt{V}_{\boldsymbol{\ell}}(\mathbf{v}) \leq \|\mathbf{v}\|_2^2\Big(\frac{\overline{\ell} + \overline{\ell}^{-1}}{4} - \frac{1}{2} + K_p\ell_1^{(2-p)}d^{\frac{2-p}{\min\{q,2\}}}\Big). \tag{D.4}$$

Note that the optimal $p$ to minimize $\epsilon_Q$ is obtained by minimizing:

$$\lambda(p) = \Big(\frac{1/p}{2/p - 1}\Big)\Big(\frac{1/p}{2/p - 1}\Big)^{1-p}\delta^{1-p}$$

where $\delta = \ell_1 d^{\frac{1}{\min\{q,2\}}}$.

Taking the first-order derivative of $\lambda(p)$, the optimal $p^*$ is given by

$$p^* = \begin{cases} \frac{\delta - 2}{\delta - 1}, & \delta \geq 2 \\ 0, & \delta < 2. \end{cases} \tag{D.5}$$

Substituting (D.5) into (D.4) gives (4.1), which completes the proof.

## E   PROOF OF THEOREM 2 (CODE-LENGTH BOUND)

Let $|\cdot|$ denote the length of a binary string. In this section, we obtain an upper bound on $\mathbb{E}_\omega \mathbb{E}_{\mathbf{q}_{\boldsymbol{\ell}}}[|\text{CODE} \circ \mathrm{Q}(Q_{\boldsymbol{\ell}}(g(\mathbf{x};\omega));\boldsymbol{\ell})|]$, *i.e.*, the expected number of communication bits per iteration. We recall from Section 3.2 that $\mathbf{v}$ is uniquely represnted by the tuple $(\|\mathbf{v}\|_q, \mathbf{s}, \mathbf{u})$. We first

encode the norm $\|\mathbf{v}\|_q$ using $C_b$ bits where. In practice, we use standard 32-bit floating point encoding.

We then use one bit to encode the sign of each nonzero entry of $\mathbf{u}$. Let $\mathcal{S}_j := \{i : u_i \in [l_j, l_{j+1}]\}$ and $N_j := |\mathcal{S}_j|$ for $j \in [s]$. We now provide the expression for probabilities associated with our symbols to be coded, *i.e.*, $\{\ell_0, \ell_1, \ldots, \ell_{s+1}\}$. The associated probabilities can computed using the weighted sum of the conditional CDFs of normalized coordinates in QAda and quantization levels:

**Proposition 2.** *Let $j \in [s]$. The probability of occurrence of $\ell_j$ (weight of symbol $\ell_j$) is given by*

$$p_j = \Pr(\ell_j) = \int_{\ell_{j-1}}^{\ell_j} \frac{u - \ell_{j-1}}{\ell_j - \ell_{j-1}} \, \mathrm{d}\tilde{F}(u) + \int_{\ell_j}^{\ell_{j+1}} \frac{\ell_{j+1} - u}{\ell_{j+1} - \ell_j} \, \mathrm{d}\tilde{F}(u)$$

*where $\tilde{F}$ is the weighted sum of the conditional CDFs of normalized coordinates in QAda. In addition, we have*

$$p_0 = \Pr(\ell_0 = 0) = \int_0^{\ell_1} \frac{1 - u}{\ell_1} \, \mathrm{d}\tilde{F}(u) \text{ and } p_{s+1} = \Pr(\ell_{s+1} = 1) = \int_{\ell_s}^1 \frac{u - \ell_s}{1 - \ell_s} \, \mathrm{d}\tilde{F}(u).$$

We have an upper bound on the expected number of nonzero entries as follows:

**Lemma 3.** *Let $\mathbf{v} \in \mathbb{R}^d$. The expected number of nonzeros in $Q_{\boldsymbol{\ell}}(\mathbf{v})$ is given by*

$$\mathbb{E}_{\mathbf{q}_{\boldsymbol{\ell}}}[\|Q_{\boldsymbol{\ell}}(\mathbf{v})\|_0] = (1 - p_0)d. \tag{E.1}$$

We then send the the associated codeword to encode each coordinate of $\mathbf{u}$. The optimal expected code-length for transmitting one random symbol is within one bit of the entropy of the source (Cover & Thomas, 2006). So the number of required information bits to transmit entries of $\mathbf{u}$ is bounded above by $d(H(L) + 1)$ where $H(L) = -\sum_{j=1}^s p_j \log(p_j)$ is the entropy in bits. Putting everything together, we have

$$\mathbb{E}_\omega \mathbb{E}_{\mathbf{q}_{\boldsymbol{\ell}}}[|\mathrm{CODE} \circ \mathrm{Q}(Q_{\boldsymbol{\ell}}(g(\mathbf{x}; \omega)); \boldsymbol{\ell})|] \leq C_b + (1 - p_0)d + (H(L) + 1)d$$

where $C_b = \mathcal{O}(1)$ is a universal constant. Finally, we note that the entropy of a source with $n$ outcomes is upper bounded by $\log(n)$.

## F    PROOF OF THEOREM 3 (Q-GENX UNDER ABSOLUTE NOISE)

We note that the output of Algorithm 1 follows the iterates of (Q-GenX):

$$X_{t+1/2} = X_t - \frac{\gamma_t}{K} \sum_{k=1}^K \hat{V}_{k,t}$$

$$Y_{t+1} = Y_t - K^{-1} \sum_{k=1}^K \hat{V}_{k,t+1/2} \tag{Q-GenX}$$

$$X_{t+1} = \gamma_{t+1} Y_{t+1}$$

We first prove the following *Template Inequality* for (Q-GenX), which is a useful milestone to prove both Theorems 3 and 4.

**Proposition 3** (Template inequality). *Let $X \in \mathbb{R}^d$. Suppose the iterates $X_t$ of (Q-GenX) are updated with some non-increasing step-size schedule $\gamma_t$ for $t = 1, 1/2, \ldots$ Then, we have*

$$\sum_{t=1}^T \left\langle \frac{1}{K} \sum_{k=1}^K \hat{V}_{k,t+1/2}, X_{t+1/2} - X \right\rangle \leq \frac{\|X\|_*^2}{2\gamma_{T+1}} + \frac{1}{2K^2} \sum_{t=1}^T \gamma_t \sum_{k=1}^K \|\hat{V}_{k,t+1/2} - \hat{V}_{k,t}\|_*^2$$

$$- \frac{1}{2} \sum_{t=1}^T \frac{1}{\gamma_t} \|X_t - X_{t+1/2}\|_*^2. \tag{F.1}$$

*Proof.* We first decompose $\frac{1}{K}\langle \sum_{k=1}^{K}\hat{V}_{k,t+1/2}, X_{t+1/2} - X\rangle$ into two terms and note:

$$\frac{1}{K}\Big\langle \sum_{k=1}^{K}\hat{V}_{k,t+1/2}, X_{t+1/2} - X\Big\rangle = S_A + S_B.$$

where

$$S_A = \frac{1}{K}\Big\langle \sum_{k=1}^{K}\hat{V}_{k,t+1/2}, X_{t+1/2} - X_{t+1}\Big\rangle$$

and

$$S_B = \frac{1}{K}\Big\langle \sum_{k=1}^{K}\hat{V}_{k,t+1/2}, X_{t+1} - X\Big\rangle.$$

Note that the update rule in (Q-GenX) implies:

$$\begin{aligned}
S_B &= \langle Y_t - Y_{t+1}, X_{t+1} - X\rangle \\
&= \Big\langle Y_t - \frac{\gamma_{t+1}}{\gamma_t}Y_{t+1}, X_{t+1} - X\Big\rangle + \Big\langle \frac{\gamma_{t+1}}{\gamma_t}Y_{t+1} - Y_{t+1}, X_{t+1} - X\Big\rangle \\
&= \frac{1}{\gamma_t}\langle \gamma_t Y_t - \gamma_{t+1}Y_{t+1}, X_{t+1} - X\rangle + \Big(\frac{1}{\gamma_{t+1}} - \frac{1}{\gamma_t}\Big)\langle -\gamma_{t+1}Y_{t+1}, X_{t+1} - X\rangle \\
&= \frac{1}{\gamma_t}\langle X_t - X_{t+1}, X_{t+1} - X\rangle + \Big(\frac{1}{\gamma_{t+1}} - \frac{1}{\gamma_t}\Big)\langle -X_{t+1}, X_{t+1} - X\rangle.
\end{aligned}$$

By algebraic manipulations, we can further show that

$$\begin{aligned}
S_B &= \frac{1}{\gamma_t}\left(\frac{1}{2}\|X_t - X\|_*^2 - \frac{1}{2}\|X_{t+1} - X\|_*^2 - \frac{1}{2}\|X_{t+1} - X_t\|_*^2\right) \\
&\quad + \Big(\frac{1}{\gamma_{t+1}} - \frac{1}{\gamma_t}\Big)\left(\frac{1}{2}\|X\|_*^2 - \frac{1}{2}\|X_{t+1}\|_*^2 - \frac{1}{2}\|X_{t+1} - X\|_*^2\right) \\
&\leq \frac{1}{2\gamma_t}\|X_t - X\|_*^2 - \frac{1}{2\gamma_{t+1}}\|X_{t+1} - X\|_*^2 - \frac{1}{2\gamma_t}\|X_t - X_{t+1}\|_*^2 + \Big(\frac{1}{2\gamma_{t+1}} - \frac{1}{2\gamma_t}\Big)\|X\|_*^2
\end{aligned}$$

where the last inequality holds by dropping $-\frac{1}{2}\|X_{t+1}\|_*^2$.

Rearranging the terms in the above expression and substituting $S_B$, we have

$$\begin{aligned}
\frac{1}{2\gamma_{t+1}}\|X_{t+1} - X\|_*^2 &\leq \frac{1}{2\gamma_t}\|X_t - X\|_*^2 - \frac{1}{2\gamma_t}\|X_t - X_{t+1}\|_*^2 + \Big(\frac{1}{2\gamma_{t+1}} - \frac{1}{2\gamma_t}\Big)\|X\|_*^2 \\
&\quad - \frac{1}{K}\Big\langle \sum_{k=1}^{K}\hat{V}_{k,t+1/2}, X_{t+1} - X\Big\rangle \\
&= \frac{1}{2\gamma_t}\|X_t - X\|_*^2 - \frac{1}{2\gamma_t}\|X_t - X_{t+1}\|_*^2 + \Big(\frac{1}{2\gamma_{t+1}} - \frac{1}{2\gamma_t}\Big)\|X\|_*^2 \\
&\quad + \frac{1}{K}\Big\langle \sum_{k=1}^{K}\hat{V}_{k,t+1/2}, X_{t+1/2} - X_{t+1}\Big\rangle - \frac{1}{K}\Big\langle \sum_{k=1}^{K}\hat{V}_{k,t+1/2}, X_{t+1/2} - X\Big\rangle.
\end{aligned}$$

$$\tag{F.2}$$

On the other hand, we have

$$\begin{aligned}
\frac{\gamma_t}{K}\Big\langle \sum_{k=1}^{K}\hat{V}_{k,t}, X_{t+1/2} - X\Big\rangle &= \langle X_t - X_{t+1/2}, X_{t+1/2} - X\rangle \\
&= \frac{1}{2}\|X_t - X\|_*^2 - \frac{1}{2}\|X_t - X_{t+1/2}\|_*^2 - \frac{1}{2}\|X_{t+1/2} - X\|_*^2
\end{aligned}$$

$$\tag{F.3}$$

Substituting $X = X_{t+1}$ and dividing both sides of (F.3) by $\gamma_t$, we have

$$\frac{1}{K}\Big\langle \sum_{k=1}^{K} \hat{V}_{k,t}, X_{t+1/2} - X_{t+1} \Big\rangle = \frac{1}{2\gamma_t}\|X_t - X_{t+1}\|_*^2 - \frac{1}{2\gamma_t}\|X_t - X_{t+1/2}\|_*^2 \\ - \frac{1}{2\gamma_t}\|X_{t+1/2} - X_{t+1}\|_*^2. \tag{F.4}$$

Combining (F.2) and (F.4), we have

$$\frac{1}{K}\Big\langle \sum_{k=1}^{K} \hat{V}_{k,t+1/2}, X_{t+1/2} - X \Big\rangle \leq \frac{1}{2\gamma_t}\|X_t - X\|_*^2 - \frac{1}{2\gamma_{t+1}}\|X_{t+1} - X\|_*^2 + \big(\frac{1}{2\gamma_{t+1}} - \frac{1}{2\gamma_t}\big)\|X\|_*^2 \\ + \frac{1}{K}\Big\langle \sum_{k=1}^{K} \hat{V}_{k,t+1/2} - \hat{V}_{k,t}, X_{t+1/2} - X_{t+1} \Big\rangle \\ - \frac{1}{2\gamma_t}\|X_t - X_{t+1/2}\|_*^2 - \frac{1}{2\gamma_t}\|X_{t+1} - X_{t+1/2}\|_*^2.$$

Summing the above for $t = 1, \ldots, T$ and telescoping, we have

$$\frac{1}{K}\sum_{t=1}^{T}\Big\langle \sum_{k=1}^{K} \hat{V}_{k,t+1/2}, X_{t+1/2} - X \Big\rangle \leq \frac{1}{2\gamma_1}\|X_1 - X\|_*^2 - \frac{1}{2\gamma_{T+1}}\|X_{T+1} - X\|_*^2 + \big(\frac{1}{2\gamma_{T+1}} - \frac{1}{2\gamma_1}\big)\|X\|_*^2 \\ + \frac{1}{K}\sum_{t=1}^{T}\Big\langle \sum_{k=1}^{K} \hat{V}_{k,t+1/2} - \hat{V}_{k,t}, X_{t+1/2} - X_{t+1} \Big\rangle \\ - \sum_{t=1}^{T}\frac{1}{2\gamma_t}\|X_t - X_{t+1/2}\|_*^2 - \sum_{t=1}^{T}\frac{1}{2\gamma_t}\|X_{t+1} - X_{t+1/2}\|_*^2.$$

Substituting $X_1 = 0$, we have[3]

$$\frac{1}{K}\sum_{t=1}^{T}\Big\langle \sum_{k=1}^{K} \hat{V}_{k,t+1/2}, X_{t+1/2} - X \Big\rangle \leq \frac{1}{2\gamma_{T+1}}\|X\|_*^2 + \frac{1}{K}\sum_{t=1}^{T}\Big\langle \sum_{k=1}^{K} \hat{V}_{k,t+1/2} - \hat{V}_{k,t}, X_{t+1/2} - X_{t+1} \Big\rangle \\ - \sum_{t=1}^{T}\frac{1}{2\gamma_t}\|X_t - X_{t+1/2}\|_*^2 - \sum_{t=1}^{T}\frac{1}{2\gamma_t}\|X_{t+1} - X_{t+1/2}\|_*^2. \tag{F.5}$$

By applying Cauchy–Schwarz and triangle inequalities, we have

$$\frac{1}{K}\Big\langle \sum_{k=1}^{K} \hat{V}_{k,t+1/2} - \hat{V}_{k,t}, X_{t+1/2} - X_{t+1} \Big\rangle \leq \sum_{k=1}^{K}\|\hat{V}_{k,t+1/2} - \hat{V}_{k,t}\|_* \|\frac{1}{K}(X_{t+1/2} - X_{t+1})\|_*. \tag{F.6}$$

Furthermore, since $ab \leq \frac{\gamma_t}{2K^2}a^2 + \frac{K^2}{2\gamma_t}b^2$, we have

$$\sum_{t=1}^{T}\frac{1}{K}\Big\langle \sum_{k=1}^{K} \hat{V}_{k,t+1/2} - \hat{V}_{k,t}, X_{t+1/2} - X_{t+1} \Big\rangle \leq \sum_{t=1}^{T}\frac{\gamma_t}{2K^2}\sum_{k=1}^{K}\|(\hat{V}_{k,t+1/2} - \hat{V}_{k,t})\|_*^2 \\ + \sum_{t=1}^{T}\frac{1}{2\gamma_t}\|X_{t+1/2} - X_{t+1}\|_*^2. \tag{F.7}$$

Substituting (F.7) into (F.5) and applying the convexity of $\|\cdot\|_*^2$, we obtain (F.1), which completes the proof. ∎

---

[3]The substitution $X_1 = 0$ is just for notation simplicity and can be relaxed at the expense of obtaining a slightly more complicated expression.

The following lemmas show how additional noise due to compression affects the upper bounds under *absolute noise* and *relative noise* models in Assumptions 2 and 3, respectively. Let $\mathbf{q}_\ell \sim \mathbb{P}_Q$ represent $d$ variables sampled i.i.d. for random quantization in Definition 1. We remind that $\mathbf{q}_\ell$ is *independent* of the random sample $\omega \sim \mathbb{P}$ in Eq. (2.1). We consider a general unbiased and normalized compression scheme, which has a bounded variance as in Theorem 1 and a bound on the expected number of communication bits to encode $Q_\ell(\mathbf{v})$, *i.e.*, the output of CODE $\circ$ Q introduced in Section 3.2, which satisfies Theorem 2. Then we have:

**Lemma 4** ( Unbiased compression under absolute noise). *Let $\mathbf{x} \in \mathbb{R}^d$ and $\omega \sim \mathbb{P}$. Suppose the oracle $g(\mathbf{x}; \omega)$ satisfies Assumption 2. Suppose $Q_\ell$ satisfies Theorems 1 and 2. Then the compressed $Q_\ell(g(\mathbf{x}; \omega))$ satisfies Assumption 2 with*

$$\mathbb{E}\left[\|Q_\ell(g(\mathbf{x}; \omega)) - A(\mathbf{x})\|_2^2\right] \leq \epsilon_Q M^2 + \sigma^2. \tag{F.8}$$

*Furthermore, the number of bits to encode $Q_\ell(g(\mathbf{x}; \omega))$ is bounded by the upper bound in (4.2).*

*Proof.* The almost sure boudedness and unbiasedness are immediately followed by the construction of the unbiased $Q_\ell$. In particular, we note that the maximum additional norm when compressing $Q_\ell(g(\mathbf{x}; \omega))$ happens when all normalized coordinates of $g(\mathbf{x}; \omega)$, which we call them $u_1, \ldots, u_d$ are mapped to the upper level $\ell_{\tau(u)+1}$ in Definition 1. The additional norm multiplier is bounded by $\sum_{i=1}^d (\ell_{\tau(u_i)+1} - u_i)^2 \leq \sum_{i=1}^d u_i^2 = 1$. Then we have $Q_\ell(g(\mathbf{x}; \omega)) \leq 2M$ a.s., so the additional upper bound is constant. The final property also holds as follows:

$$\begin{aligned}
\mathbb{E}_\omega \mathbb{E}_{\mathbf{q}_\ell}\left[\|Q_\ell(g(\mathbf{x}; \omega)) - A(\mathbf{x})\|_2^2\right] &= \mathbb{E}_\omega \mathbb{E}_{\mathbf{q}_\ell}\left[\|Q_\ell(g(\mathbf{x}; \omega)) \pm g(\mathbf{x}; \omega) - A(\mathbf{x})\|_2^2\right] \\
&= \mathbb{E}_\omega \mathbb{E}_{\mathbf{q}_\ell}\left[\|Q_\ell(g(\mathbf{x}; \omega)) - g(\mathbf{x}; \omega)\|_2^2\right] + \mathbb{E}_\omega\left[\|U(\mathbf{x}; \omega)\|_2^2\right] \\
&\leq \epsilon_Q \mathbb{E}_\omega\left[\|g(\mathbf{x}; \omega)\|_2^2\right] + \sigma^2 \\
&\leq \epsilon_Q M^2 + \sigma^2
\end{aligned}$$

where the second step holds due to the unbiasedness of $\mathbf{q}_\ell$ and the last inequality holds since $\|g(\mathbf{x}; \omega)\|_* \leq M$ a.s. ∎

**Lemma 5** (Unbiased compression under relative noise). *Let $\mathbf{x} \in \mathbb{R}^d$ and $\omega \sim \mathbb{P}$. Suppose the oracle $g(\mathbf{x}; \omega)$ satisfies Assumption 3. Suppose $Q_\ell$ satisfies Theorems 1 and 2. Then the compressed $Q_\ell(g(\mathbf{x}; \omega))$ satisfies Assumption 3 with*

$$\mathbb{E}\left[\|Q_\ell(g(\mathbf{x}; \omega)) - A(\mathbf{x})\|_2^2\right] \leq \left(\epsilon_Q(c+1) + c\right)\|A(\mathbf{x})\|_2^2. \tag{F.9}$$

*Furthermore, the number of bits to encode $Q_\ell(g(\mathbf{x}; \omega))$ is bounded by the upper bound in (4.2).*

*Proof.* The almost sure boudedness and unbiasedness are immediately followed by the construction of the unbiased $Q_\ell$. The final property also holds as follows:

$$\begin{aligned}
\mathbb{E}_\omega \mathbb{E}_{\mathbf{q}_\ell}\left[\|Q_\ell(g(\mathbf{x}; \omega)) - A(\mathbf{x})\|_2^2\right] &= \mathbb{E}_\omega \mathbb{E}_{\mathbf{q}_\ell}\left[\|Q_\ell(g(\mathbf{x}; \omega)) - g(\mathbf{x}; \omega)\|_2^2\right] + \mathbb{E}_\omega\left[\|U(\mathbf{x}; \omega)\|_2^2\right] \\
&\leq \epsilon_Q \mathbb{E}_\omega\left[\|g(\mathbf{x}; \omega)\|_2^2\right] + c\|A(\mathbf{x})\|_2^2 \\
&= \epsilon_Q \mathbb{E}_\omega\left[\|U(\mathbf{x}; \omega) + A(\mathbf{x})\|_2^2\right] + c\|A(\mathbf{x})\|_2^2 \\
&= \epsilon_Q\left(\mathbb{E}_\omega\left[\|U(\mathbf{x}; \omega)\|_2^2\right] + \|A(\mathbf{x})\|_2^2\right) + c\|A(\mathbf{x})\|_2^2 \\
&\leq \left(\epsilon_Q(c+1) + c\right)\|A(\mathbf{x})\|_2^2.
\end{aligned}$$

where the first and fourth steps hold due to unbiasedness of $\mathbf{q}_\ell$ and the noise model, respectively. ∎

We now prove our main theorme:

**Theorem 5** (Q-GenX under absolute noise). *Let $\mathcal{C} \subset \mathbb{R}^d$ denote a compact neighborhood of a solution for (VI) and let $D^2 := \sup_{X \in \mathcal{C}} \|X - X_0\|^2$. Suppose that the oracle and the problem (VI) satisfy Assumptions 1 and 2, respectively, Algorithm 1 is executed for $T$ iterations on $K$ processors with an adaptive step-size $\gamma_t = K(1 + \sum_{i=1}^{t-1} \sum_{k=1}^K \|\hat{V}_{k,i} - \hat{V}_{k,i+1/2}\|^2)^{-1/2}$, and quantization levels are updated $J$ times where $\ell_j$ with variance bound $\epsilon_{Q,j}$ in (4.1) and code-length bound $N_{Q,j}$ in (4.2) is used for $T_j$ iterations with $\sum_{j=1}^J T_j = T$. Then we have*

$$\mathbb{E}\left[\text{Gap}_\mathcal{C}\left(\frac{1}{T}\sum_{t=1}^T X_{t+1/2}\right)\right] = \mathcal{O}\left(\frac{(\sum_{j=1}^J \sqrt{\epsilon_{Q,j} T_j / T} M + \sigma) D^2}{\sqrt{TK}}\right).$$

In addition, Algorithm 1 requires each processor to send at most $\frac{2}{T}\sum_{j=1}^{J} T_j N_{Q,j}$ communication bits per iteration in expectation.

In the following, we prove the results using the template inequality Proposition 3 and noise anslysis in Lemma 4.

As a preliminary step, we prove this proposition for an *adaptive step-size* with a *non-adaptive $Q_\ell$*, which satisfies Theorem 1.

**Proposition 4** (Algorithm 1 under absolute noise and fixed compression scheme). *Under the setup described in Theorem 3, with an adaptive step-size $\gamma_t = K(1+\sum_{i=1}^{t-1}\sum_{k=1}^{K}\|\hat{V}_{k,i}-\hat{V}_{k,i+1/2}\|^2)^{-1/2}$ and non-adaptive $Q_\ell$ satisfying Theorem 1, we have*

$$\mathbb{E}\Big[\operatorname{Gap}_{\mathcal{C}}\Big(\frac{1}{T}\sum_{t=1}^{T} X_{t+1/2}\Big)\Big] = \mathcal{O}\Big(\frac{(\sqrt{\epsilon_Q}M+\sigma)D^2}{\sqrt{TK}}\Big).$$

*Proof.* Suppose that we do not apply compression, *i.e.*, $\epsilon_Q = 0$. By the template inequality Proposition 3, we have

$$\sum_{t=1}^{T}\Big\langle \frac{1}{K}\sum_{k=1}^{K}\hat{V}_{k,t+1/2}, X_{t+1/2}-X\Big\rangle \leq \frac{\|X\|_*^2}{2\gamma_{T+1}} + \frac{1}{2K^2}\sum_{t=1}^{T}\gamma_t\sum_{k=1}^{K}\|\hat{V}_{k,t+1/2}-\hat{V}_{k,t}\|_*^2. \quad \text{(F.10)}$$

Let denote the LHS and RHS of (F.10) by $S_A = \sum_{t=1}^{T}\Big\langle \frac{1}{K}\sum_{k=1}^{K}\hat{V}_{k,t+1/2}, X_{t+1/2}-X\Big\rangle$ and $S_B = \frac{\|X\|_*^2}{2\gamma_{T+1}} + \frac{1}{2K^2}\sum_{t=1}^{T}\gamma_t\sum_{k=1}^{K}\|\hat{V}_{k,t+1/2}-\hat{V}_{k,t}\|_*^2$, respectively. Then, by the noise model (2.1) and monotonicity of operator $A$, we have

$$\begin{aligned}
S_A &= \sum_{t=1}^{T}\Big\langle \frac{1}{K}\sum_{k=1}^{K}A_k(X_{t+1/2}), X_{t+1/2}-X\Big\rangle + \sum_{t=1}^{T}\Big\langle \frac{1}{K}\sum_{k=1}^{K}U_{k,t+1/2}, X_{t+1/2}-X\Big\rangle \\
&\geq \sum_{t=1}^{T}\Big\langle \frac{1}{K}\sum_{k=1}^{K}A_k(X), X_{t+1/2}-X\Big\rangle + \sum_{t=1}^{T}\Big\langle \frac{1}{K}\sum_{k=1}^{K}U_{k,t+1/2}, X_{t+1/2}-X\Big\rangle \\
&= \frac{T}{K}\sum_{k=1}^{K}\langle A_k(X), \overline{X}_{T+1/2}-X\rangle + \sum_{t=1}^{T}\Big\langle \frac{1}{K}\sum_{k=1}^{K}U_{k,t+1/2}, X_{t+1/2}-X\Big\rangle
\end{aligned}$$

where $A_k = A$ for $k \in [K]$. Therefore, by rearranging the terms Eq. (F.10) using above inequality, we have

$$\begin{aligned}
\frac{T}{K}\sum_{k=1}^{K}\langle A_k(X), \overline{X}_{T+1/2}-X\rangle &\leq -\sum_{t=1}^{T}\langle \frac{1}{K}\sum_{k=1}^{K}U_{k,t+1/2}, X_{t+1/2}-X\rangle \\
&\quad + \frac{\|X\|_*^2}{2\gamma_{T+1}} + \frac{1}{2K^2}\sum_{t=1}^{T}\gamma_t\sum_{k=1}^{K}\|\hat{V}_{k,t+1/2}-\hat{V}_{k,t}\|_*^2.
\end{aligned} \quad \text{(F.11)}$$

By taking supermom on both sides of Eq. (F.11), dividing by $T$, and taking expectation, we have

$$\mathbb{E}\Big[\frac{1}{K}\sum_{k=1}^{K}\sup_X \langle A_k(X), \overline{X}_{T+1/2}-X\rangle\Big] \leq \frac{1}{T}(S_1 + S_2 + S_3) \quad \text{(F.12)}$$

where $S_1 = \mathbb{E}\Big[\frac{D^2}{2\gamma_{T+1}}\Big]$, $S_2 = \mathbb{E}\Big[\frac{1}{2K^2}\sum_{t=1}^{T}\gamma_t\sum_{k=1}^{K}\|\hat{V}_{k,t+1/2}-\hat{V}_{k,t}\|_*^2\Big]$, and $S_3 = \mathbb{E}\Big[\sup_X\sum_{t=1}^{T}\langle\frac{1}{K}\sum_{k=1}^{K}U_{k,t+1/2}, X_{t+1/2}-X\rangle\Big]$. We now bound $S_1$, $S_2$, and $S_3$ from above,

individually. For $S_1$, we have

$$
\begin{aligned}
S_1 &= \mathbb{E}\left[\frac{D^2}{2\gamma_{T+1}}\right] \\
&= \frac{D^2}{2K}\mathbb{E}\left[\sqrt{1+\sum_{i=1}^{T-1}\sum_{k=1}^{K}\|\hat{V}_{k,i}-\hat{V}_{k,i+1/2}\|^2}\right] \\
&\leq \frac{D^2}{2K}\sqrt{1+\sum_{i=1}^{T-1}\sum_{k=1}^{K}\mathbb{E}\left[\|\hat{V}_{k,i}-\hat{V}_{k,i+1/2}\|^2\right]} \\
&\leq \frac{D^2}{2K}\sqrt{1+\sum_{i=1}^{T-1}\sum_{k=1}^{K}2(\mathbb{E}[\|\hat{V}_{k,i}\|^2]+\mathbb{E}[\|\hat{V}_{k,i+1/2}\|^2])} \\
&\leq \frac{D^2}{2K}\sqrt{1+4KT\sigma^2}.
\end{aligned}
\tag{F.13}
$$

We also have

$$
\begin{aligned}
S_2 &= \mathbb{E}\left[\frac{1}{2K^2}\sum_{t=1}^{T}\gamma_t\sum_{k=1}^{K}\|\hat{V}_{k,t+1/2}-\hat{V}_{k,t}\|_*^2\right] \\
&= \frac{1}{2}\mathbb{E}\left[\sum_{t=1}^{T}\left(\frac{\gamma_t}{K^2}-\frac{\gamma_{t+1}}{K^2}\right)\sum_{k=1}^{K}\|\hat{V}_{k,t+1/2}-\hat{V}_{k,t}\|_*^2\right]+\frac{1}{2}\mathbb{E}\left[\sum_{t=1}^{T}\frac{\gamma_{t+1}}{K^2}\sum_{k=1}^{K}\|\hat{V}_{k,t+1/2}-\hat{V}_{k,t}\|_*^2\right] \\
&\leq 2\mathbb{E}\left[\sum_{t=1}^{T}\left(\frac{\gamma_t}{K^2}-\frac{\gamma_{t+1}}{K^2}\right)K\sigma^2\right]+\frac{1}{2}\mathbb{E}\left[\sum_{t=1}^{T}\frac{\gamma_{t+1}}{K^2}\sum_{k=1}^{K}\|\hat{V}_{k,t+1/2}-\hat{V}_{k,t}\|_*^2\right] \\
&\leq 2\sigma^2+\frac{1}{2K}\mathbb{E}\left[\sum_{t=1}^{T}\frac{\sum_{k=1}^{K}\|\hat{V}_{k,t+1/2}-\hat{V}_{k,t}\|_*^2}{\sqrt{1+\sum_{t=1}^{T}\sum_{k=1}^{K}\|\hat{V}_{k,t+1/2}-\hat{V}_{k,t}\|_*^2}}\right] \\
&\leq 2\sigma^2+\frac{1}{2K}\mathbb{E}\left[\sqrt{1+\sum_{t=1}^{T}\sum_{k=1}^{K}\|\hat{V}_{k,t+1/2}-\hat{V}_{k,t}\|_*^2}\right] \\
&\leq 2\sigma^2+\frac{1}{2K}\sqrt{1+\sum_{t=1}^{T}\sum_{k=1}^{K}\mathbb{E}\left[\|\hat{V}_{k,t+1/2}-\hat{V}_{k,t}\|_*^2\right]} \\
&\leq 2\sigma^2+\frac{1}{2K}\sqrt{1+4\sigma^2KT}.
\end{aligned}
\tag{F.14}
$$

Finally we note that

$$
\begin{aligned}
S_3 &= \mathbb{E}\left[\sup_{X}\sum_{t=1}^{T}\langle\frac{1}{K}\sum_{k=1}^{K}U_{k,t+1/2},X_{t+1/2}-X\rangle\right] \\
&= \mathbb{E}\left[\sup_{X}\sum_{t=1}^{T}\langle\frac{1}{K}\sum_{k=1}^{K}U_{k,t+1/2},X\rangle\right]-\mathbb{E}\left[\sup_{X}\sum_{t=1}^{T}\langle\frac{1}{K}\sum_{k=1}^{K}U_{k,t+1/2},X_{t+1/2}\rangle\right].
\end{aligned}
\tag{F.15}
$$

We bound the first term in the RHS of (F.15) using the following known lemma:

**Lemma 6** (Bach & Levy 2019). *Let $\mathcal{C}\in\mathbb{R}^d$ be a convex set and $h:\mathcal{C}\to\mathbb{R}$ be a 1-strongly convex w.r.t. a norm $\|\cdot\|$. Assume that $h(\mathbf{x})-\min_{\mathbf{x}\in\mathcal{C}}h(\mathbf{x})\leq D^2/2$ for all $\mathbf{x}\in\mathcal{C}$. Then, for any martingale difference $(\mathbf{z}_t)_{t=1}^{T}\in\mathbb{R}^d$ and any $\mathbf{x}\in\mathcal{C}$, we have*

$$
\mathbb{E}\left[\left\langle\sum_{t=1}^{T}\mathbf{z}_t,\mathbf{x}\right\rangle\right]\leq\frac{D^2}{2}\sqrt{\sum_{t=1}^{T}\mathbb{E}[\|\mathbf{z}_t\|^2]}.
$$

Using Lemma 6, the first term in the RHS of (F.15) is bounded by

$$\frac{1}{K}\mathbb{E}\left[\sup_X \sum_{t=1}^{T}\langle \sum_{k=1}^{K} U_{k,t+1/2}, X\rangle\right] \leq \frac{D^2}{2K}\sqrt{\mathbb{E}\left[\sum_{t=1}^{T}\sum_{k=1}^{K}\|U_{k,t+1/2}\|^2\right]}$$
$$\leq \frac{D^2\sigma\sqrt{T}}{2\sqrt{K}}. \tag{F.16}$$

Similarly, we can bound the second term in the RHS of (F.15). Combining the results in Eq. (F.13), Eq. (F.14), and Eq. (F.16) and applying Lemma 4, we obtain the upper bound in Proposition 4. ∎

Applying Proposition 4 with the scaled step size schedule in Theorem 3, we complete the proof for an adaptive compression scheme along the lines of (Faghri et al., 2020, Theorem 4).

## G  PROOF OF THEOREM 4 (Q-GENX UNDER RELATIVE NOISE)

We first remind the theorem statement:

**Theorem 6** (Q-GenX under relative noise). *Let $\mathcal{C} \subset \mathbb{R}^d$ denote a compact neighborhood of a solution for (VI) and let $D^2 := \sup_{X \in \mathcal{C}} \|X - X_0\|^2$. Suppose that the oracle and the problem (VI) satisfy Assumptions 1, 3, and 4, Algorithm 1 is executed for $T$ iterations on $K$ processors with an adaptive step-size $\gamma_t = K(1 + \sum_{i=1}^{t-1}\sum_{k=1}^{K}\|\hat{V}_{k,i} - \hat{V}_{k,i+1/2}\|^2)^{-1/2}$, and quantization levels are updated $J$ times where $\ell_j$ with variance bound $\epsilon_{Q,j}$ in (4.1) and code-length bound $N_{Q,j}$ in (4.2) is used for $T_j$ iterations with $\sum_{j=1}^{J} T_j = T$. Then we have*

$$\mathbb{E}\left[\operatorname{Gap}_{\mathcal{C}}\left(\frac{1}{T}\sum_{t=1}^{T}X_{t+1/2}\right)\right] = \mathcal{O}\left(\frac{((c+1)\sum_{j=1}^{J}T_j\epsilon_{Q,j}/T + c)D^2}{KT}\right).$$

*In addition, Algorithm 1 requires each processor to send at most $\frac{2}{T}\sum_{j=1}^{J}T_j N_{Q,j}$ communication bits per iteration in expectation.*

Suppose that we do not apply compression, *i.e.*, $\epsilon_Q = 0$. We first remind the template inequality in (F.1), which holds for any $\gamma_t$ and noise model:

$$\sum_{t=1}^{T}\left\langle\frac{1}{K}\sum_{k=1}^{K}\hat{V}_{k,t+1/2}, X_{t+1/2} - X\right\rangle \leq \frac{\|X\|_*^2}{2\gamma_{T+1}} + \frac{1}{2K^2}\sum_{t=1}^{T}\gamma_t\sum_{k=1}^{K}\|\hat{V}_{k,t+1/2} - \hat{V}_{k,t}\|_*^2$$
$$- \frac{1}{2}\sum_{t=1}^{T}\frac{1}{\gamma_t}\|X_t - X_{t+1/2}\|_*^2.$$

In the following proposition, we show that under $\gamma_t$ and relative noise model in Theorem 4, $\sum_{t=1}^{T}\mathbb{E}\left[\|A(X_{t+1/2})\|_*^2 + \|A(X_t)\|_*^2\right]$ is summable in the sense that $\sum_{t=1}^{T}\mathbb{E}\left[\|A(X_{t+1/2})\|_*^2 + \|A(X_t)\|_*^2\right] = \mathcal{O}(1/\gamma_T)$.

**Proposition 5** (Sum operator output under relative noise). *Let $X^*$ denote a solution of (VI). Under the setup described in Theorem 4, we have:*

$$\sum_{t=1}^{T}\mathbb{E}\left[\|A(X_{t+1/2})\|_*^2 + \|A(X_t)\|_*^2\right] \leq \mathbb{E}\left[\frac{\|X^*\|_*^2}{2\gamma_{T+1}}\right]. \tag{G.1}$$

*Proof.* Substituting $X = X^*$ into $\mathbb{E}\big[\big\langle \frac{1}{K}\sum_{k=1}^{K}\hat{V}_{k,t+1/2}, X_{t+1/2} - X\big\rangle\big]$ and applying the law of total expectation, we have:

$$
\begin{aligned}
\mathbb{E}\Big[\Big\langle \frac{1}{K}\sum_{k=1}^{K}\hat{V}_{k,t+1/2}, X_{t+1/2} - X^*\Big\rangle\Big] &= \mathbb{E}\Big[\frac{1}{K}\sum_{k=1}^{K}\mathbb{E}[\langle \hat{V}_{k,t+1/2}, X_{t+1/2} - X^*\rangle | X_{t+1/2}]\Big] \\
&= \mathbb{E}\Big[\frac{1}{K}\sum_{k=1}^{K}\langle A_k(X_{t+1/2}), X_{t+1/2} - X^*\rangle\Big] \\
&= \mathbb{E}\Big[\langle A(X_{t+1/2}), X_{t+1/2} - X^*\rangle\Big] \\
&\geq \mathbb{E}\Big[\langle A(X_{t+1/2}) - A(X^*), X_{t+1/2} - X^*\rangle\Big] \\
&\geq \beta\mathbb{E}[\|A(X_{t+1/2})\|_*^2]
\end{aligned}
\tag{G.2}
$$

where the fourth and fifth inequalities hold due to the definition of the monotone operator Eq. (VI) and $\beta$-cocoecivity in Eq. (4.3), respectively.

Applying the lower bound in Eq. (G.2) into Eq. (F.1), we obtain:

$$
\begin{aligned}
\sum_{t=1}^{T}\beta\mathbb{E}[\|A(X_{t+1/2})\|_*^2] \leq \mathbb{E}\Big[&\frac{\|X^*\|_*^2}{2\gamma_{T+1}} + \frac{1}{2K^2}\sum_{t=1}^{T}\gamma_t\sum_{k=1}^{K}\|\hat{V}_{k,t+1/2} - \hat{V}_{k,t}\|_*^2 \\
&- \frac{1}{2}\sum_{t=1}^{T}\frac{1}{\gamma_t}\|X_t - X_{t+1/2}\|_*^2\Big].
\end{aligned}
\tag{G.3}
$$

Moreover, by lower bounding (LHS) of the above:

$$
\sum_{t=1}^{T}\mathbb{E}[\|A(X_{t+1/2})\|_*^2] = \sum_{t=1}^{T}\mathbb{E}[1/K\sum_{k=1}^{K}\|A(X_{t+1/2})\|^2]
\tag{G.4}
$$

$$
\geq \frac{1}{c}\sum_{t=1}^{T}\mathbb{E}[1/K\sum_{k=1}^{K}\|\hat{V}_{k,t+1/2}\|_*^2]
\tag{G.5}
$$

with the second inequality being obtained by the relative noise condition. On the other hand, applying Cauchy–Schwarz and $\beta$-cocoecivity in Eq. (4.3) imply $\|X_t - X_{t+1/2}\|_*^2 \geq \beta^2\|A(X_t) - A(X_{t+1/2})\|_*^2$. It follows that:

$$
\begin{aligned}
\frac{1}{2}\mathbb{E}\Big[&\sum_{t=1}^{T}\beta\|A(X_{t+1/2})\|_*^2 + \sum_{t=1}^{T}\frac{1}{\gamma_t}\|X_t - X_{t+1/2}\|_*^2\Big] \\
&\geq \frac{1}{2}\mathbb{E}\Big[\sum_{t=1}^{T}\beta\|A(X_{t+1/2})\|_*^2 + \sum_{t=1}^{T}\frac{\beta^2}{\gamma_t}\|A(X_t) - A(X_{t+1/2})\|_*^2\Big] \\
&\geq \frac{1}{2}\mathbb{E}\Big[\sum_{t=1}^{T}\beta\|A(X_{t+1/2})\|_*^2 + \sum_{t=1}^{T}\frac{\beta^2}{\gamma_t}\|A(X_t) - A(X_{t+1/2})\|_*^2\Big] \\
&\geq \frac{1}{2}\min\Big\{\beta, \frac{\beta^2}{\gamma_0}\Big\}\sum_{t=1}^{T}\mathbb{E}\Big[\|A(X_{t+1/2})\|_*^2 + \|A(X_t) - A(X_{t+1/2})\|_*^2\Big] \\
&\geq \frac{1}{2}\min\Big\{\beta, \frac{\beta^2}{\gamma_0}\Big\}\sum_{t=1}^{T}\mathbb{E}\Big[\|A(X_t)\|_*^2\Big] \\
&\geq \frac{1}{2}\min\Big\{\beta, \frac{\beta^2}{\gamma_0}\Big\}\sum_{t=1}^{T}\mathbb{E}\Big[1/K\sum_{k=1}^{K}\|A(X_t)\|_*^2\Big] \\
&\geq \frac{1}{2c}\min\Big\{\beta, \frac{\beta^2}{\gamma_0}\Big\}\sum_{t=1}^{T}\mathbb{E}\Big[1/K\sum_{k=1}^{K}\|\hat{V}_{k,t}\|_*^2\Big]
\end{aligned}
\tag{G.6}
$$

where the last inequality holds due to the relative noise condition. Combining the above inequalities we get the following:

$$\frac{\beta}{c}\sum_{t=1}^{T}\mathbb{E}[1/K\sum_{k=1}^{K}\|\hat{V}_{k,t+1/2}\|_*^2] \leq \mathbb{E}[\frac{\|X\|_*^2}{2\gamma_{T+1}} + \frac{1}{2K^2}\sum_{t=1}^{T}\gamma_t\sum_{k=1}^{K}\|\hat{V}_{k,t+1/2} - \hat{V}_{k,t}\|_*^2] \qquad \text{(G.7)}$$

and

$$\frac{1}{2c}\min\left\{\beta, \frac{\beta^2}{\gamma_0}\right\}\sum_{t=1}^{T}\mathbb{E}\left[1/K\sum_{k=1}^{K}\|\hat{V}_{k,t}\|_*^2\right] \leq \mathbb{E}[\frac{\|X\|_*^2}{2\gamma_{T+1}} + \frac{1}{2K^2}\sum_{t=1}^{T}\gamma_t\sum_{k=1}^{K}\|\hat{V}_{k,t+1/2} - \hat{V}_{k,t}\|_*^2]$$
$$\text{(G.8)}$$

Therefore, by adding the above inequalities we get:

$$\frac{\beta}{c}\sum_{t=1}^{T}\mathbb{E}[1/K\sum_{k=1}^{K}\|\hat{V}_{k,t+1/2}\|_*^2] + \frac{1}{2c}\min\left\{\beta, \frac{\beta^2}{\gamma_0}\right\}\sum_{t=1}^{T}\mathbb{E}\left[1/K\sum_{k=1}^{K}\|\hat{V}_{k,t}\|_*^2\right]$$
$$\text{(G.9)}$$
$$\leq 2\mathbb{E}\left[\frac{\|X\|_*^2}{2\gamma_{T+1}} + \frac{1}{2K^2}\sum_{t=1}^{T}\gamma_t\sum_{k=1}^{K}\|\hat{V}_{k,t+1/2} - \hat{V}_{k,t}\|_*^2\right]$$

We now establish an upper bound on the R.H.S. of Eq. (G.9). We first note that:

$$\mathbb{E}\left[\frac{1}{2K^2}\sum_{t=1}^{T}\gamma_t\sum_{k=1}^{K}\|\hat{V}_{k,t+1/2} - \hat{V}_{k,t}\|_*^2\right]$$

$$= \mathbb{E}\left[\frac{1}{2K^2}\sum_{t=1}^{T}(\gamma_t - \gamma_{t+1})\sum_{k=1}^{K}\|\hat{V}_{k,t+1/2} - \hat{V}_{k,t}\|_*^2 + \frac{1}{2K^2}\sum_{t=1}^{T}\gamma_{t+1}\sum_{k=1}^{K}\|\hat{V}_{k,t+1/2} - \hat{V}_{k,t}\|_*^2\right]$$

$$\leq \mathbb{E}\left[\frac{1}{2K}(4M^2)\sum_{t=1}^{T}(\gamma_t - \gamma_{t+1}) + \frac{1}{2K^2}\sum_{t=1}^{T}\gamma_{t+1}\sum_{k=1}^{K}\|\hat{V}_{k,t+1/2} - \hat{V}_{k,t}\|_*^2\right]$$

$$\leq \mathbb{E}\left[\frac{2M^2}{K}\gamma_1 + \frac{1}{2K^2}\sum_{t=1}^{T}\gamma_{t+1}\sum_{k=1}^{K}\|\hat{V}_{k,t+1/2} - \hat{V}_{k,t}\|_*^2\right]$$

$$\leq \mathbb{E}\left[2M^2 + \frac{1}{2K^2}\sum_{t=1}^{T}\gamma_{t+1}\sum_{k=1}^{K}\|\hat{V}_{k,t+1/2} - \hat{V}_{k,t}\|_*^2\right]$$

$$\leq \mathbb{E}\left[2M^2 + \frac{1}{K}\sqrt{1 + \sum_{t=1}^{T}\sum_{k=1}^{K}\|\hat{V}_{k,t+1/2} - \hat{V}_{k,t}\|_*^2}\right]$$

$$\lesssim \mathbb{E}[\frac{1}{\gamma_{T+1}}].$$
$$\text{(G.10)}$$

Therefore, an upper bound on the R.H.S. of Eq. (G.9) is given by :

$$\frac{\beta}{c}\sum_{t=1}^{T}\mathbb{E}[1/K\sum_{k=1}^{K}\|\hat{V}_{k,t+1/2}\|_*^2] + \frac{1}{2c}\min\left\{\beta, \frac{\beta^2}{\gamma_0}\right\}\sum_{t=1}^{T}\mathbb{E}\left[1/K\sum_{k=1}^{K}\|\hat{V}_{k,t}\|_*^2\right] \leq \mathbb{E}\left[\frac{\|X^*\|_*^2 + 1}{\gamma_{T+1}}\right].$$
$$\text{(G.11)}$$

■

To establish a lower bound on the L.H.S. of Eq. (G.9), we first note that:

$$\mathbb{E}\left[\frac{1}{K^2}\sum_{t=1}^{T}\sum_{k=1}^{K}\|\hat{V}_{k,t+1/2} - \hat{V}_{k,t}\|_*^2\right] = \mathbb{E}\left[\frac{1}{K^2}\left(1 + \sum_{t=1}^{T}\sum_{k=1}^{K}\|\hat{V}_{k,t+1/2} - \hat{V}_{k,t}\|_*^2\right)\right] - \frac{1}{K^2}$$
$$\text{(G.12)}$$
$$= \mathbb{E}[\frac{1}{\gamma_{T+1}^2}] - \frac{1}{K^2}.$$

We also note that:

$$\frac{K}{4c} \min\left\{\beta, \frac{\beta^2}{\gamma_0}\right\} \mathbb{E}\big[(1/K^2 \sum_{t=1}^{T} \sum_{k=1}^{K} \|\hat{V}_{k,t+1/2} - \hat{V}_{k,t}\|_*^2)\big]$$
$$\leq \frac{\beta}{2c} \sum_{t=1}^{T} \mathbb{E}[1/K \sum_{k=1}^{K} \|\hat{V}_{k,t+1/2}\|_*^2] + \frac{1}{2c} \min\left\{\beta, \frac{\beta^2}{\gamma_0}\right\} \sum_{t=1}^{T} \mathbb{E}\Big[1/K \sum_{k=1}^{K} \|\hat{V}_{k,t}\|_*^2\Big]. \tag{G.13}$$

Hence, combining, Eqs. (G.11)–(G.13), we can find an upper bound on $\mathbb{E}[\frac{1}{\gamma_{T+1}^2}]$:

$$\frac{K}{4c} \min\left\{\beta, \frac{\beta^2}{\gamma_0}\right\} \mathbb{E}[\frac{1}{\gamma_{T+1}^2}] \leq \big[\|X^*\|^2 + 1\big] \mathbb{E}[\frac{1}{\gamma_{T+1}}]$$
$$= \big[\|X^*\|^2 + 1\big] \mathbb{E}[\sqrt{\frac{1}{\gamma_{T+1}^2}}]$$
$$\leq \big[\|X^*\|^2 + 1\big] \sqrt{\mathbb{E}[\frac{1}{\gamma_{T+1}^2}]}$$

with the last inequality being obtained by Jensen's inequality. So we have

$$\mathbb{E}[\frac{1}{\gamma_{T+1}}] \leq \frac{4c}{K} \max\left\{\frac{1}{\beta}, \frac{\gamma_0}{\beta^2}\right\}. \tag{G.14}$$

Similar to the proof of Theorem 3, we have

$$\mathbb{E}\Big[\sup_X \langle A(X), \overline{X}_{T+1/2} - X\rangle\Big] \leq \frac{1}{T}(S_1 + S_2 + S_3) \tag{G.15}$$

where $S_1 = \mathbb{E}\Big[\frac{D^2}{2\gamma_{T+1}}\Big]$, $S_2 = \mathbb{E}\Big[\frac{1}{2K^2} \sum_{t=1}^{T} \gamma_t \sum_{k=1}^{K} \|\hat{V}_{k,t+1/2} - \hat{V}_{k,t}\|_*^2\Big]$, and $S_3 = \mathbb{E}\Big[\sup_X \sum_{t=1}^{T} \langle \frac{1}{K} \sum_{k=1}^{K} U_{k,t+1/2}, X - X_{t+1/2}\rangle\Big]$.

By (G.10), we have

$$\mathbb{E}\left[\frac{1}{2K^2} \sum_{t=1}^{T} \gamma_t \sum_{k=1}^{K} \|\hat{V}_{k,t+1/2} - \hat{V}_{k,t}\|_*^2\right] \lesssim \mathbb{E}\left[\frac{1}{\gamma_{T+1}}\right]. \tag{G.16}$$

We now decompose $S_3$ into two terms $S_3 = \mathbb{E}\Big[\sup_X \sum_{t=1}^{T} \langle \frac{1}{K} \sum_{k=1}^{K} U_{k,t+1/2}, X\rangle\Big] - \mathbb{E}\Big[\sum_{t=1}^{T} \langle \frac{1}{K} \sum_{k=1}^{K} U_{k,t+1/2}, X_{t+1/2}\rangle\Big]$.

Let the supremum on the first terms is attained by $X = X^o$. We can also establish an upper bound on the first term using Lemma 6:

$$\frac{1}{K}\mathbb{E}\left[\sup_X \sum_{t=1}^T \sum_{k=1}^K \langle \sum U_{k,t+1/2}, X\rangle\right] = \frac{1}{K}\mathbb{E}\left[\langle \sum_{t=1}^T \sum_{k=1}^K U_{k,t+1/2}, X^o\rangle\right]$$

$$= \frac{D^2}{2K}\sqrt{\mathbb{E}\left[\|\sum_{t=1}^T \sum_{k=1}^K U_{k,t+1/2}\|_*^2\right]}$$

$$\leq \frac{D^2}{2\sqrt{K}}\sqrt{\mathbb{E}\left[\sum_{t=1}^T c\|A(X_{t+1/2})\|_*^2\right]}$$

$$\leq \frac{D^2}{2\sqrt{K}}\sqrt{c\mathbb{E}\left[\frac{\|X^*\|_*^2}{2\gamma_{T+1}}\right]}$$

(G.17)

where the last inequality holds by Proposition 5.

Finally, by the law of total expectation, we have:

$$\mathbb{E}\left[\sum_{t=1}^T \langle \sum_{k=1}^K U_{k,t+1/2}, X_{t+1/2}\rangle\right] = \mathbb{E}\left[\sum_{t=1}^T \sum_{k=1}^K \mathbb{E}[\langle U_{k,t+1/2}, X_{t+1/2}\rangle | X_{t+1/2}]\right] = 0. \quad \text{(G.18)}$$

Substituting Eq. (G.14), Eq. (G.17), and Eq. (G.18) into Eq. (G.15), we have

$$\mathbb{E}[\text{Gap}_{\mathcal{C}} \overline{X}_{T+1/2}] = \mathcal{O}\left(\frac{1}{T}\mathbb{E}\left[\frac{D^2}{2\gamma_{T+1}}\right]\right). \quad \text{(G.19)}$$

Following similar analysis as in Proposition 4 and applying Lemma 5 with the scaled step size schedule in Theorem 4, we complete the proof for an adaptive compression scheme along the lines of (Faghri et al., 2020, Theorem 4).

## H EXPERIMENTAL DETAILS AND ADDITIONAL EXPERIMENTS

In order to validate our theoretical results, we build on the code base of Gidel et al. (2019) and run an instantiation of Q-GenX obtained by combining ExtraAdam with the compression offered by the `torch_cgx` pytorch extension of Markov et al. (2022), and train a WGAN-GP (Arjovsky et al., 2017) on CIFAR10 (Krizhevsky, 2009).

Since `torch_cgx` uses OpenMPI (Gabriel et al., 2004) as its communication backend, we use OpenMPI as the communication backend for the full gradient as well for a fairer comparison. We deliberately do *not* tune any hyperparameters to fit the larger batchsize since simliar to (Gidel et al., 2019), we do not claim to set a new SOTA with these experiments but simply want to show that our theory holds up in practice and can potentially lead to improvements. For this, we present a basic experiment showing that even for a *very small problem size* and *a heuristic base compression method of cgx*, we can achieve a *noticeable speedup* of around $8\%$. We expect further gains to be achievable for larger problems and more advanced compression methods. Given differences in terms of settings and the *lack of any code, let alone an efficient implementations that can be used in a real-world setting (i.e. CUDA kernels integrated with networking)*, it is difficult to impossible to conduct a fair comparison with Beznosikov et al. (2021).

We follow exactly the setup of (Gidel et al., 2019) except that we share an effective batch size of 1024 across 3 nodes (strong scaling) connected via Ethernet, and use Layernorm (Ba et al., 2016) instead of Batchnorm (Ioffe & Szegedy, 2015) since Batchnorm is known to be challenging to work with in distributed training as well as interacting badly with the WGAN-GP penalty. The results are shown in Fig. 2a, showing evolution of FID and Fig. 2b showing the accumulated total time spent backpropagating. We note that we do *not* scale the learning rate or any other hyperparameters to account for these two changes so this experiment is *not* meant to claim SOTA performance, merely to illustrate that

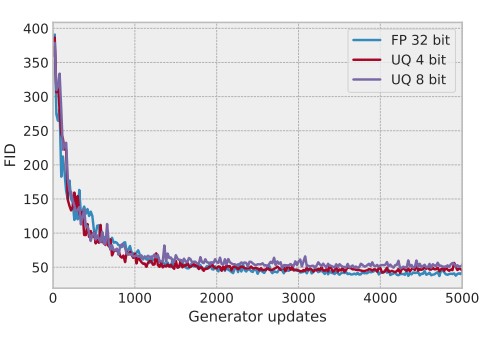

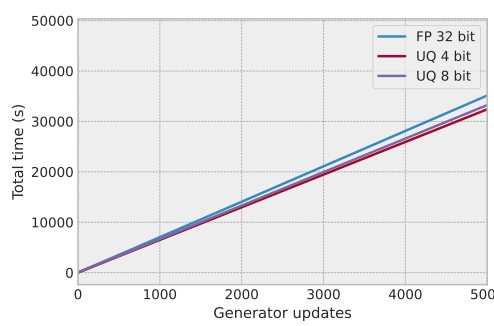

**(a)** FID evolution during training

**(b)** Total time spent on backpropagation and gradient exchanges

**Figure 2:** Comparing full gradient ExtraAdam with a simple instantiation of QGenX. FID stands for Frechet inception distance, which is a standard GAN quality metric introduced in (Heusel et al., 2017).

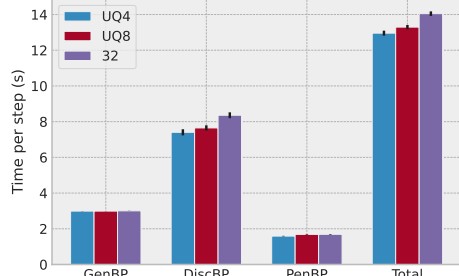

| Mode | time (s) | | | |
| | GenBP | DiscBP | PenBP | Total |
| --- | --- | --- | --- | --- |
| UQ4 | 2.99 | 7.40 | 1.59 | 12.96 |
| UQ8 | 2.99 | 7.65 | 1.69 | 13.29 |
| FP32 | 3.00 | 8.36 | 1.69 | 14.05 |

**Figure 3:** Fine grained comparison of average `.backward()` times on generator, discriminator, gradient penalty as well as total training time. The `.backward()` function is where pytorch DDP handles gradient exchange

**Figure 4:** Comparing Q-GenX with QSGDA

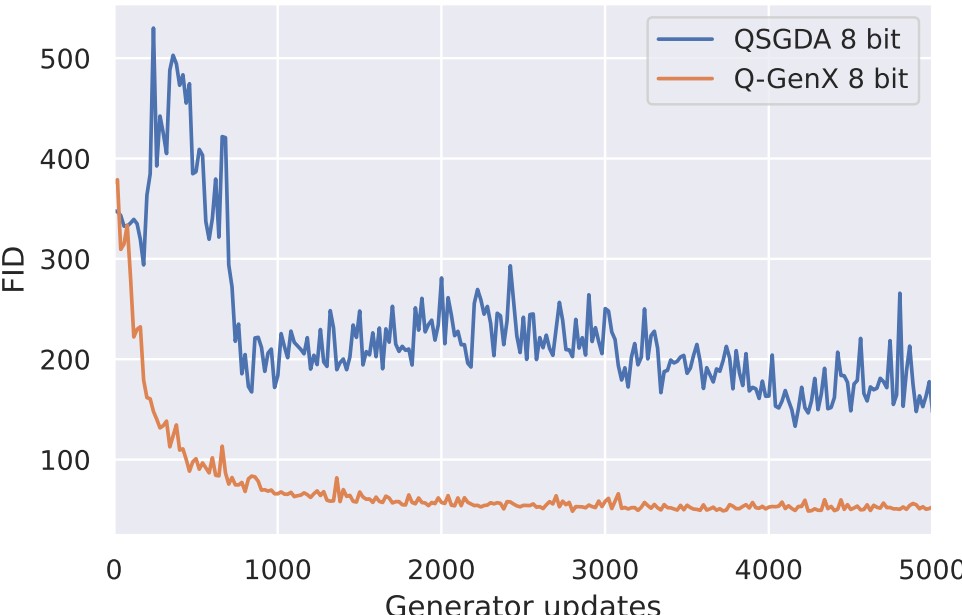

1. Even with the simplest possible unbiased quantization on a relatively small-scale setup, we can observe speedup (about $10\%$).

2. This speedup does not drastically change the performance.

We compare training using the full gradient of 32 bit (FP32) to training with gradients compressed to 8 (UQ8) and 4 bits (UQ4) using a bucket size of 1024. Figure 3 shows a more fine grained breakdown of the time used for back propagation (BP) where the network activity takes place. GenBP, DiscBP and PenBP refer to the backpropagation for generator, discriminator and the calculation of the gradient penalty, respectively. Total refers to the sum of these times.

We used Weights and Biases (Biewald, 2020) for all experiment tracking. Our time measurements are performed with pythons `time.time()` function which has microsecond precision on Linux, measuring only backward propagation times and total training time, excluding plotting, logging etc. The experiments were performed on 3 Nvidia V100 GPUs (1 per node) using a Kubernetes cluster and an image built on the `torch_cgx` Docker image.

## H.1 COMPARISON WITH QSGDA

Figure 4 compares Q-GenX with QSGDA of Beznosikov et al. (2022), the only method without explicit variance reduction. Due to the extra-gradient template, Q-GenX is able to make steady progress without variance reduction.

## I TRADE-OFF BETWEEN NUMBER OF ITERATIONS AND TIME PER ITERATION

In this section, we build on our theoretical results in Theorems 3 and 4 to capture the trade-off between the number of iterations to converge and time per iteration, which includes total time required to update a model on each GPU.

Using the results of Theorems 3 and 4, we can obtain the minimum number of iterations required to *guarantee an expected gap of $\epsilon$*, which is a measure of solution quality. In particular, under absolute noise model and average gradient variance bound $\bar{\epsilon}_Q = \sum_{j=1}^{J} T_j \epsilon_{Q,j}/T$, the minimum number of iterations to guarantee an expected gap of $\epsilon$ is $T(\epsilon, \bar{\epsilon}_Q) = (\bar{\epsilon}_Q M^2 + \sigma^2)^2 D^4/\epsilon^2$. Now suppose that the time per iteration, which includes overall computation, encoding, and communication times to compute, compress, send, receive, decompress and update one iteration, is denoted by $\Delta$. Decreasing the number of communication bits, i.e., compressing more aggressively increases the sufficient number of iterations $T(\epsilon, \bar{\epsilon}_Q)$ and decreases time per iteration, $\Delta$, due to communication savings, which captures the trade-off. Theoretically, the best compression method is the one with the minimum overall wall-clock time bounded by $T(\epsilon, \bar{\epsilon}_Q)\Delta$. The exact optimal point depends on the specific problem to solve (dataset, loss, etc.) and the hyperparameters chosen (architecture, number of bits, etc.), which together determine $\bar{\epsilon}_Q$ and the implementation details of the algorithm, networking, and compression, along with the cluster setup and hyperparameters, which together influence $\Delta$. We defer more refined analysis of the optimal point to future work.

## J    EXAMPLES MOTIVATING ASSUMPTION 3

In this section, we provide some popular examples which motivate Assumption 3:

**Example J.1** (Random coordinate descent (RCD)). Consider a smooth convex function $f$ over $\mathbb{R}^d$. At iteration $t$, the RCD algorithm draws one coordinate $i_t \in [d]$ uniformly at random and computes the partial derivative $v_{i,t} = \partial f/\partial x_{i_t}$. Subsequently, the $i$-th derivative is updated as $X_{i,t+1} = X_{i,t} - d \cdot \alpha \cdot v_{i,t}$ where $\alpha > 0$ denotes the step-size.

This update rule can be written in an abstract recursive form as $\mathbf{x}^+ = \mathbf{x} - \alpha g(\mathbf{x}; \mu)$ where $g_i(\mathbf{x}; \mu) = d \cdot \partial f/\partial x_i \cdot \mu$ and $\mu$ is drawn uniformly at random from the set of basis vectors $\{\mathbf{e}_1, \ldots, \mathbf{e}_d\} \subseteq \mathbb{R}^d$. We note that $\mathbb{E}[g(\mathbf{x}; \mu)] = \nabla f(\mathbf{x})$. Furthermore, since $\partial f/\partial x_i = 0$ at the minima of $f$, we also have $g(\mathbf{x}^*; \mu) = 0$ if $\mathbf{x}^*$ is a minimizer of $f$, *i.e.*, the variance of the random vector $g(\mathbf{x}; \mu)$ vanishes at the minima of $f$. It is not difficult to show that $\mathbb{E}_\mu \|g(\mathbf{x}; \mu) - \nabla f(\mathbf{x})\|^2 = \mathcal{O}(\|\nabla f(x)\|^2)$, which satisfies Assumption 3 with $A = \nabla f$.

**Example J.2** (Random player updating). Consider an $N$-player convex game with loss functions $f_i$, $i \in [N]$. Suppose, at each stage, player $i$ is selected with probability $p_i$ to play an action following its individual gradient descent rule $X_{i,t+1} = X_{i,t} + \gamma_t/p_i V_{i,t}$ where $V_{i,t} = \nabla_i f_i(X_t)$ denotes player $i$'s individual gradient at the state $X_t = (X_{1,t}, ..., X_{N,t})$ and $p_i$ is included for scaling reasons.

Note that $\mathbb{E}[V_t] = A(X_t)$ where $A_i(x) = \nabla_i f_i(x)$ for $i \in [N]$. It is not difficult to show that $V_t$ is an *unbiased oracle* for $A$, and since all individual components of $A$ vanish at the game's Nash equilibria, it is also straightforward to verify that $V_t$ satisfies Assumption 3.

## K    ENCODING

To further reduce communication costs, we can apply information-theoretically inspired coding schemes on top of quantization. In this section, we provide an overview of our coding schemes along the lines of (Alistarh et al., 2017; Faghri et al., 2020; Ramezani-Kebrya et al., 2021). Let $q \in \mathbb{Z}_+$. We first note that a vector $\mathbf{v} \in \mathbb{R}^d$ can be *uniquely* represented by a tuple $(\|\mathbf{v}\|_q, \mathbf{s}, \mathbf{u})$ where $\|\mathbf{v}\|_q$ is the $L^q$ norm of $\mathbf{v}$, $\mathbf{s} := [\mathrm{sgn}(v_1), \ldots, \mathrm{sgn}(v_d)]^\top$ consists of signs of the coordinates $v_i$'s, and $\mathbf{u} := [u_1, \ldots, u_d]^\top$ with $u_i = |v_i|/\|\mathbf{v}\|_q$ are the normalized coordinates. Note that $0 \leq u_i \leq 1$ for all $i \in [d]$. We define a random quantization function as follows:

**Definition 2** (Random quantization function). *Let $s \in \mathbb{Z}_+$ denote the number of quantization levels. Let $u \in [0,1]$ and $\boldsymbol{\ell} = (\ell_0, \ldots, \ell_{s+1})$ denote a sequence of $s$ quantization levels with $0 = \ell_0 < \ell_1 < \cdots < \ell_s < \ell_{s+1} = 1$. Let $\tau(u)$ denote the index of a level such that $\ell_{\tau(u)} \leq u < \ell_{\tau(u)+1}$. Let $\xi(u) = (u - \ell_{\tau(u)})/(\ell_{\tau(u)+1} - \ell_{\tau(u)})$ be the relative distance of $u$ to level $\tau(u) + 1$. We define the random function $q_{\boldsymbol{\ell}}(u) : [0,1] \to \{\ell_0, \ldots, \ell_{s+1}\}$ such that $q_{\boldsymbol{\ell}}(u) = \ell_{\tau(u)}$ with probability $1 - \xi(u)$ and $q_{\boldsymbol{\ell}}(u) = \ell_{\tau(u)+1}$ with probability $\xi(u)$. Let $q \in \mathbb{Z}_+$ and $\mathbf{v} \in \mathbb{R}^d$. We define the random quantization of $\mathbf{v}$ as follows:*

$$Q_{\boldsymbol{\ell}}(\mathbf{v}) := \|\mathbf{v}\|_q \cdot \mathbf{s} \odot [q_{\boldsymbol{\ell}}(u_1), \ldots, q_{\boldsymbol{\ell}}(u_d)]^\top$$

*where $\odot$ denotes the element-wise (Hadamard) product.*

We note that $\mathbf{q}_{\boldsymbol{\ell}} = \{q_{\boldsymbol{\ell}}(u_i)\}_{i=1,\ldots,d}$ are *independent* random variables. The encoding $\text{CODE} \circ \text{Q}(\|\mathbf{v}\|_q, \mathbf{s}, \mathbf{q}_{\boldsymbol{\ell}}) : \mathbb{R}_+ \times \{\pm 1\}^d \times \{\ell_0, \ldots, \ell_{s+1}\}^d \to \{0,1\}^*$ uses a standard floating point encoding with $C_b$ bits to represent the positive scalar $\|\mathbf{v}\|_q$, encodes the sign of each coordinate with one bit, and finally applies an *integer* encoding scheme $\Psi : \{\ell_0, \ell_1, \ldots, \ell_{s+1}\} \to \{0,1\}^*$ to *efficiently* encode each quantized normalized coordinate $q_{\boldsymbol{\ell}}(u_i)$ with the *minimum* expected code-length. Depending on how much knowledge of the distribution of the discrete alphabet of levels is known, a particular lossless prefix code can be used to encode $\mathbf{q}_{\boldsymbol{\ell}}$. In particular, if the distribution of the frequency of the discrete alphabet $\{\ell_0, \ell_1, \ldots, \ell_{s+1}\}$ is unknown but it is known that smaller values are more frequent than larger values, Elias recursive coding (ERC) can be used (Elias, 1975). ERC is a universal lossless integer coding scheme with a recursive and efficient encoding and decoding schemes, which assigns shorter codes to smaller values. If the distribution of the frequency of the discrete alphabet $\{\ell_0, \ell_1, \ldots, \ell_{s+1}\}$ is known or can be estimated efficiently, we use Huffman coding, which has an efficient encoding/decoding scheme and achieves the *minimum expected code-length* among methods encoding symbols separately (Cover & Thomas, 2006).

The decoding $\text{DEQ} \circ \text{CODE} : \{0,1\}^* \to \mathbb{R}^d$ in Algorithm 1 first reads $C_b$ bits to reconstruct $\|\mathbf{v}\|_q$. Then it applies $\Psi^{-1} : \{0,1\}^* \to \{\ell_0, \ell_1, \ldots, \ell_{s+1}\}$ to decode the index of the first coordinate, depending on whether the decoded entry is zero or nonzero, it may read one bit indicating the sign, and then proceeds to decode its value. It then decodes the next symbol. The decoding continues mimicking the encoding scheme and finishes when all quantized coordinates are decoded. Note that the decoding will fully recover $Q_{\boldsymbol{\ell}}(\mathbf{v})$ because the coding scheme is lossless. One may slightly improve the coding efficiency in terms of the expected code-length by encoding blocks of symbols at the cost of increasing encoding/decoding complexity. We focus on lossless prefix coding schemes, which encode symbols separately due to their encoding/decoding simplicity (Cover & Thomas, 2006).

To implement an efficient Huffman code, we need to estimate probabilities w.r.t. the symbols in our discrete alphabet $\{\ell_0, \ell_1, \ldots, \ell_{s+1}\}$. This discrete distribution can be estimated by properly estimating the marginal probability density function (PDF) of normalized coordinates along the lines of *e.g.,* (Faghri et al., 2020, Proposition 6).

Given quantization levels $\boldsymbol{\ell}_t$ and the marginal PDF of normalized coordinates, $K$ processors can construct the Huffman tree in parallel. A Huffman tree of a source with $s + 2$ symbols can be constructed in time $O(s)$ through sorting the symbols by the associated probabilities. It is well-known that Huffman codes minimize the expected code-length:

**Theorem 7** (Cover & Thomas 2006, Theorems 5.4.1 and 5.8.1)**.** *Let $Z$ denote a random source with a discrete alphabet $\mathcal{Z}$. The expected code-length of an optimal prefix code to compress $Z$ is bounded by $H(Z) \leq \mathbb{E}[L] \leq H(Z) + 1$ where $H(Z) \leq \log_2(|\mathcal{Z}|)$ is the entropy of $Z$ in bits.*

