# OpenReview forum: "Distributed Extra-gradient with Optimal Complexity and Communication Guarantees"
_ICLR.cc/2023/Conference — ICLR 2023 poster_

### Official Review · Reviewer_PUDg · 2022-10-21

**Confidence:** 2
**Correctness:** 4
**Technical Novelty And Significance:** 2
**Empirical Novelty And Significance:** Not applicable
**Recommendation:** 6

**Clarity, Quality, Novelty And Reproducibility:**

The writing is extremely clear, but the contributions of this work relative to
others could be more explicit. The experiments in the appendix are minimal but
appear to be reproducible.

**Strength And Weaknesses:**

**Strengths:**
- This paper is very well written and provides a comprehensive survey of
  related works.
- The Q-GenX algorithm is indeed practical, and the authors provide strong
  theoretical guarantees about its performance.

**Weaknesses:**
- The exact contributions of this work over previous methods (aside from the
  unification aspect) should be made more clear. The discussion after Theorem 5
  talks about more processors leading to faster convergence, but how does this
  convergence rate compare to existing methods for different slices of the VI
  problem (e.g., convex, saddle-points)?
- Given the nature of this paper, there should be some experiments in the main
  body. Some GAN experiments are discussed in Appendix H, but it seems
  incomplete.

**Summary Of The Paper:**

This work studies distributed (synchronous) algorithms for monotone variational
inequality (VI) optimization, generalizing convex minimization, saddle-point
problems, and games. Specifically, it proposes a quantized, general
extra-gradient framework called "Q-GenX" that unifies previous methods.  This
approach uses unbiased and adaptive compression techniques for communicating
local stochastic gradients, and it uses an adaptive learning rate rule for
which the authors prove convergence guarantees: $O(1/T)$ under relative nosie
and $O(1/\sqrt{T})$ in the presence of absolute noise. For this VI problem,
they show that increasing the number of processes can accelerate convergence.
Lastly, they train GANs on multiple GPUs in the appendix to support their
theoretical results.



**Summary Of The Review:**

This work is polished and well-organized, but the comparison to prior works and
experiments to support the Q-GenX algorithm seem to be lacking. The algorithm
is clean and practical, and come with strong theoretical guarantees. This paper
could be accepted to ICLR, but it would benefit from addressing the weaknesses
pointed out above.

---

> ### Author Response · Authors · 2022-11-10
> **Response to Reviewer PUDg**
>
> We thank the reviewer for their thoughtful comments which we address one by one below.
>
> ------
>
> -  Compare convergence rates with existing rate for convex, saddle-points problems:
>
> Compared to saddle point problems, our rates are optimal. This can be verified by lower bounds in (Beznosikov et al., 2020). For convex problems in deterministic settings, the rate can be improved to ${\cal O}(1/T^2)$ via acceleration. However, it is known that in the stochastic and distributed settings, our rates cannot be improved even with acceleration. E.g., for convex and smooth problems under absolute noise model, the lower bound of $\Omega (\frac{1}{\sqrt{TK}})$ based on our notation can be established by (Woodworth et al., 2021)[Theorem 1]  and setting the number of gradients per round to one.
>
> In the revision, we have clarified this in Section 4.
>
> Aleksandr Beznosikov, Valentin Samokhin, and Alexander Gasnikov. "Distributed saddle-point problems: Lower bounds, optimal and robust algorithms." arXiv preprint arXiv:2010.13112, 2020.
>
>
> Blake E. Woodworth, Brian Bullins, Ohad Shamir, and Nathan Srebro. "The min-max complexity of distributed stochastic convex optimization with intermittent communication." In Conference on Learning Theory (COLT), 2021.
>
>
> ------
>
> -   Move experiments to the main draft:
>
> In the revision, we have moved experiments to the main draft and numerically compared Q-GenX with QSGDA of Beznosikov et al. (2022) [here](https://imgur.com/a/73F2ZPE).  This is the only communication-efficient VI solver **without variance reduction in the stochastic setting** that is proposed in the literature, which can be compared fairly using our distributed framework.
>
> Aleksandr Beznosikov, Eduard Gorbunov, Hugo Berard, and Nicolas Loizou. "Stochastic gradient descent-ascent: Unified theory and new efficient methods." arXiv preprint arXiv:2202.07262, 2022.
>
>
> ------

---

> ### Author Response · Authors · 2022-11-15
> **Follow-up**
>
> Dear reviewer,
>
> We are currently unaware of the extent to which our responses have clarified your concerns, and willing to provide further clarifications to you.
>
> Regards,
> Authors

---

> ### Comment · Reviewer_PUDg · 2022-11-28
> **Followup response**
>
> Thank you to the authors for their careful responses to all of the reviewers. I have increased my score from 5 --> 6.

---

> > ### Author Response · Authors · 2022-11-28
> > **Thank you for constructive comments and re-evaluating our work**
> >
> > We thank the reviewer for carefully reading all of the reviews and increasing the score.

---

### Official Review · Reviewer_4SD9 · 2022-10-26

**Confidence:** 2
**Correctness:** 4
**Technical Novelty And Significance:** 3
**Empirical Novelty And Significance:** 3
**Recommendation:** 8

**Clarity, Quality, Novelty And Reproducibility:**

Clarity and quality:

Overall the paper is clearly written. The paper is technical and it is possible that I misunderstood part of it. However, the authors did a good job explaining all the material.

Section 3.3. (optimization of the quantization levels) could be clearer, as well as the difference between encoding and quantization.

Novelty:

This papers seems to be the first to consider communication efficient variant of EG. The rates are reasonable and recover known results as special case.

No reproducibility issues.

**Strength And Weaknesses:**

Strength

- The compression part of the paper is precise and specific. The compression operators are described and the number of bits communicating is computed. I am not sure, but these calculations seem to be new. So there is also a contribution here, additional to the optimization part (analysis of EG)

- The optimization part (EG algo) seems equally solid. Convergence is in restricted gap and recover the deterministic result as a special case.

-The assumptions on the noise cover several types of stochastic gradients

-The meta algorithm actually covers several algorithms for monotone VI


Weaknesses

-In many applications of ML, monotonicity does not hold (Monotonicity is like convexity). This paper deals only with the monotone case

**Summary Of The Paper:**

This paper considers a monotone variational inequality to be solved in a Federated learning setup, with a central server and multiple clients. They develop a version of the extra gradient (EG) algorithm in this setting where the bottleneck is the communication: In their algorithm, communications between the central server and the clients are compressed.

The resulting algorithm is a version of EG with compression and they show convergence rates in terms of the restricted gap function under different assumptions on the noise in the stochastic "gradient". The stronger assumption on the noise ensures zero noise at the optimum and therefore recover the deterministic rate of EG. Besides, the authors compute the communication cost and show reduction in number of bits communicated at each round.

**Summary Of The Review:**

A solid work that considers an idealized setting (monotonicity) but provides precise and competitive guarantees for both the computation complexity and the communication complexity of the proposed algorithm.

---

> ### Author Response · Authors · 2022-11-10
> **Response to Reviewer 4SD9**
>
> We thank the reviewer for their thoughtful comments which we address one by one below.
>
> ------
>
> - Clarify optimization of quantization levels and the difference between encoding and quantization:
>
> In the revision, we have improved the presentation of Section 3. In particular, in section 3.3, we added “Instead of using a heuristically chosen sequence of quantization levels,  adaptive quantization estimates distribution of uncompressed original vectors, i.e.,  dual vectors, by computing sufficient statistics of a parametric distribution, optimizes quantization levels to minimize the quantization error, and updates those levels adaptively throughout the course of training as the distribution changes.” We have also clarified that marginal cumulative distribution functions of normalized coordinates are tailored based on statistical properties of “original stochastic dual vectors” observed for example when training a GAN. A more fine-grained analysis can be done by considering two sequences of adaptive levels one for $V_{k,t}$’s and another one for $V_{k,t+1/2}$’s. While in this paper, we quantize both sequences with the same quantization scheme, it is possible to further reduce quantization errors by considering two fine-grained quantization schemes at the cost of additional computations at processors.
>
>
> In section 3.2, we have clarified that while the main focus of this paper is on quantization, we can still apply information-theoretically coding schemes *on top of quantization* to further reduce communication costs. In particular, Quantization maps real values to a discrete set of quantization levels. Then a coding scheme maps those discrete values to a bit string. Our ENCODE in Algorithm 1 of the original version is a composition of both quantization and coding, which outputs a bit string. To avoid confusion, in the revised version, we have used notations $\mathrm{CODE\circ Q}$ and $\mathrm{DEQ\circ CODE}$ in Algorithm 1,  and moved unnecessarily parts of Section 3.2 to the appendix.
>
>
> ------

---

> > ### Comment · Reviewer_4SD9 · 2022-11-14
> > **Thanks for the answer**
> >
> > OK, noted. Thanks for the clarification

---

### Official Review · Reviewer_Qv34 · 2022-10-28

**Confidence:** 4
**Clarity, Quality, Novelty And Reproducibility:** Please see the above review for furth…
**Correctness:** 3
**Technical Novelty And Significance:** 3
**Empirical Novelty And Significance:** 2
**Recommendation:** 5

**Strength And Weaknesses:**

The paper has interesting results, however, it has several issues in terms of presentation that do not allow me to suggest acceptance at this stage. Let me provide more details below.

1. The paper claims that it solves general monotone problems while truly focusing on a much smaller class of VIPs for which the analysis is typically much easier. This is the class of co-coercive VIPs.

2. On top of the co-coercivity condition, the authors mentioned "Moreover, in order to avoid trivialities, we make the following mild assumption" to refer to assumption 1. However, Assumption 1 assumes that the problem has a unique solution as well. Thus it is far from general monotone or even general co-coercive.

3. In the expression of the co-coercive condition the dual norm is used without further explanation. Note that the standard co-coercivity holds for euclidean norms.

4. For co-coercive problems assuming bounded variance, bounded noise or relative noise (growth condition) are unnecessary and quite strong assumptions. Please check the results of paper [1] where these conditions have not been used.
The authors spend half a page providing examples where Ass. 2 and 3 are satisfied which is good for pedagogical reasons but the examples are standard for anyone familiar with noise assumptions in stochastic algorithms. This space could have been used for the numerical experiment (which is not part of the main paper)

5. Proposition 1 is a well-known result. As presented now it gives the impression that this is a contribution of the authors. The authors mention: "Mathematically speaking, we have the following proposition:"

6. Presentation of Section 3 is confusing as it involves quantities that appear later in the paper. What is $Q_{\ell_t}$ what is $V_{k,t}$, etc. A good idea for clarity will be to introduce a notation paragraph at the beginning of that section 3. Note that definition 1 only appears in section 3.2 while it should have been presented before section 3.1

7. In section 4 a proof sketch is provided in the main paper which takes half a page. This space could have been used for the experiments which as it is mentioned in several parts of this work is one of its main contributions. At the moment there are no experiments in the main paper.

8. Main Issue:
Even if one checks the experiments in the appendix (not a requirement for the reviewer) the paper has no comparison with closely related works that use compression on top of distributed algorithms for solving VIPs. The authors mentioned the works of Beznosikov et al. (2021); Kovalev et al. (2022) and say that they are concurrent works. This is not true. Beznosikov et al. (2021) for example, is 1-year-old paper.

In the numerical evaluation, a proper comparison with algorithms from the above papers should be presented.

A highly related paper, that also focuses on co-coercive VIPs, and to the best of my knowledge has state-of-the-art performance in terms of distributed methods with compressors for VIPs is paper [2]. This is also non-concurrent work as it appears some time ago. A formal comparison with the distributed methods with compressed communications presented in that work is highly recommended.

Missing references:

[1] Loizou, Nicolas, Hugo Berard, Gauthier Gidel, Ioannis Mitliagkas, and Simon Lacoste-Julien. "Stochastic gradient descent-ascent and consensus optimization for smooth games: Convergence analysis under expected co-coercivity." Advances in Neural Information Processing Systems 34 (2021): 19095-19108.

[2] Beznosikov, Aleksandr, Eduard Gorbunov, Hugo Berard, and Nicolas Loizou. "Stochastic gradient descent-ascent: Unified theory and new efficient methods." arXiv preprint arXiv:2202.07262 (2022).

**Summary Of The Paper:**

The paper focuses on co-coercive variational inequalities and proposes a quantized generalized extra-gradient (Q-GenX) for solving these problems in multi-GPU settings where multiple processors/workers/clients have access to local stochastic dual vectors. In terms of theory, the paper proves $O(1/T)$ convergence of Q-GenX under relative noise and $O(1/\sqrt{T})$ under absolute noise.

**Summary Of The Review:**

As I mentioned in my review the idea of the paper is interesting. However, there are many issues in terms of presentation and numerical evaluation.

In my opinion, a fair score for the current version of the paper is "3: reject, not good enough" or "5: marginally below the acceptance threshold".

At this point, I give a score of 5 to this work.

---

> ### Author Response · Authors · 2022-11-10
> **Response to Reviewer Qv34 Part 1**
>
> We thank the reviewer for their thoughtful comments which we address one by one below.
>
> ------
>
> -  The paper claims general VIPs while analyses co-coercive VIPs:
>
> One important objective of our  work is to  try to answer the following research question for distributed settings: “whether it is possible to design a class of distributed  methods that are capable of adapting to different  oracle structures, and  achieve order-optimal rates without prior knowledge of the problem’s parameters?”
>
> In order to answer this affirmatively, we use the co-coercivity condition following the line of work of (Lin et al., 2020) done for the non-distributed framework . In that sense, co-coercivity is required to achieve that our method is agnostic relative to the  oracle’s model.
>  Important note here is that our order-optimal rate of ${\cal O} (1/\sqrt{T})$ under absolute noise does not require co-cocercivity, and relies on standard Lipschitz continuity of the respective operator. In particular, our general template inequality in Proposition 3 holds under any noise model and any non-increasing step-size schedule in a general distributed setting , which **does not require co-coercivity** and is the main building block of both Theorems 3 and 4  in the revision.
>
> Moreover, our adaptive step-size also does not depend on the noise model or co-coercivity, providing a unified universal and robust performance for a whole family of popular  algorithmic schemes .  We have clarified this in Section 4 of the revision.
>
> Tianyi Lin, Zhengyuan Zhou, Panayotis Mertikopoulos, and Michael Jordan. "Finite-time last-iterate convergence for multi-agent learning in games." In ICML 2020.
>
> ------
>
> -  For co-coercive problems absolute noise or relative noise can be relaxed (Loizou et al., 2021):
>
> Another important aspect of our work is to answer in a concrete manner the following question:
> “Are there any conditions for the method’s oracle that would close the stochastic-deterministic
> convergence gap of $1/\sqrt{T}$ and $1/T$?”
>
>  To that end, besides the “folk” absolute noise model, we built our analysis around the so-called relative noise model; firstly introduced by Polyak (1987). In ML,  this noise model has also been studied in the context of overparametrization (Oymak et al., 2020), representation learning (Zhang et al., 2021), and multi-agent learning (Lin et al., 2020). Finally, this oracle model has also been studied under the umbrella of multiplicative noise (Iusem et al., 2019) or growth conditions (Schmidt et al., 2013, Vaswani et al., 2019, Xie et al., 2020), and it is known to improve the convergence rate of stochastic gradient algorithms with non-adaptive step-sizes, even in non-smooth problems (Facchinei and Pang, 2003).
>
> Therefore, we strongly believe that this oracle’s structure provides a common and solid baseline with the SOTA literature mentioned above. As we already mentioned, co-coercivity is required for establishing universal theoretical guarantees for different oracle’s structures.
>
>  On the other hand,  Loizou et al. (2021) require quasi-strongly monotonicity to relax bounded variance/growth condition to expected co-coercivity assumption, which is not required in our setting. Please note that that the analysis in (Loizou et al., 2021) does not include an *adaptive step-size*.
>
> While our analysis is built on standard absolute/relative noise models  (Polyak, 1987, Nesterov, 2004; Nemirovski et al., 2009; Juditsky et al., 2011; Levy et al., 2018; Kavis et al., 2019; Bach & Levy, 2019; Antonakopoulos & Mertikopoulos, 2021) , which help us to accommodate an adaptive step-size without requiring quasi-strongly monotonicity, we have cited (Loizou et al., 2021) in Section 6 as an interesting future direction to study monotone VIs under expected co-coercivity and adaptive step-sizes.
>
> Mark Schmidt and Nicolas Le Roux. Fast convergence of stochastic gradient descent under a strong growth condition. arXiv:1308.6370, 2013.
>
> Sharan Vaswani, Francis Bach, and Mark Schmidt. Fast and faster convergence of SGD for over-parameterized models and an accelerated perceptron. In AISTATS 2019.
>
> Alfredo N Iusem, Alejandro Jofre, Roberto I Oliveira, and Philip Thompson. Variance-based extragradient methods with line search for stochastic variational inequalities. SIAM Journal on Optimization, 29(1): 175–206, 2019.
>
> Samet Oymak and Mahdi Soltanolkotabi. Towards moderate overparameterization: global convergence guarantees for training shallow neural networks. IEEE Journal on Selected Areas in Information Theory 2020.
>
> Yuege Xie, Xiaoxia Wu, and Rachel Ward. Linear convergence of adaptive stochastic gradient descent. In AISTATS 2020.
>
> Chiyuan Zhang, Samy Bengio, Moritz Hardt, Benjamin Recht, and Oriol Vinyals. "Understanding deep learning (still) requires rethinking generalization." Communications of the ACM 2021.

---

> > ### Author Response · Authors · 2022-12-06
> > **Happy to provide further clarification**
> >
> > We hope that we have resolved the major concerns from your side. If possible, can you please rate the paper more positively? We would be happy to provide further clarifications if needed.

---

> ### Author Response · Authors · 2022-11-10
> **Response to Reviewer Qv34 Part 2**
>
>
> ------
>
> -  Uniqueness of the solution:
>
> We thank the reviewer for pointing this out. This is actually a typo as one may easily verify from our analysis that this uniqueness assumption is not needed. We have fixed this in the revised version.
>
> ------
>
> -  Dual vs Euclidean norm:
>
> Note that the elements of $A({\bf x})$ generally lie on the so-called dual space. Therefore,  we adopted the mathematically accurate dual norm formulation. For the particular Euclidean space, i.e., $\mathbb{R}^d$ equipped with the Euclidean norm which the reviewer mentions, the induced Hilbertian structure allows us to identify the respective dual norm with its primal counterpart. That being said, if the reviewer feels that this does not serve the clarity of our paper, we are willing to update the revised version accordingly.
>
> ------
>
> -  Proposition 1 is a well-known result:
>
> The justification of the gap function as a performance criterion for the broader context of VIs is not new, as we explicitly referred the reader to the appropriate citations below Prop. 1. For the sake of completeness and clarity a formal mathematical statement is necessary. In the revision, we have changed it to **Proposition 1 (Nesterov, 2009).**
>
> ------
>
> -  Improve Presentation of Section 3:
>
> We use $Q_{\ell_t}$ to denote a random and adaptive quantization function where the quantization levels $\ell_t$ may change over time. $V_{k,t’}$ is the original (uncompressed) stochastic dual vector computed by process $k$ at time $t'$. In the revision, we have moved Definition 1 and added a notation paragraph in the beginning of Section 3.1. Besides, we extensively revised Section 3 to improve its clarity.
>
> ------
>
> -  Move experiments to the main draft:
>
> In the revision, we have removed the proof sketch and moved experiments to the main draft.
>
> ------
>
> -  Numerical comparison with (Beznosikov et al., 2021, 2022):
>
> As for comparing with the MASHA variants in Beznosikov et al. (2021), the authors did not provide the code used in their evaluation. Since their algorithm differs from most other compressed distributed algorithm in requiring  two or three randomized exchanges (send Q_dev to server,  send Q_serv +shared random bit to devices, based on this random bit,  update the variance reduction) and efficient implementation requires CUDA and C++ implementation of the networking and compression algorithms as was done in the CGX framework we use for our work, we do not think a fair comparison is realistically possible given constraints on time and manpower. If the reviewers are aware of a validated PyTorch implementation of the algorithms, we are happy to evaluate them.
>
> In the revision, we have cited and compared with QSGDA of Beznosikov et al. (2022) [here](https://imgur.com/a/73F2ZPE). This is the only communication-efficient VI solver **without variance reduction in the stochastic setting** that is proposed in the literature, which can be compared fairly using our distributed framework.
>
> Aleksandr Beznosikov, Eduard Gorbunov, Hugo Berard, and Nicolas Loizou. "Stochastic gradient descent-ascent: Unified theory and new efficient methods." arXiv preprint arXiv:2202.07262, 2022.
>
> ------

---

> ### Author Response · Authors · 2022-11-15
> **Follow-up**
>
> Dear reviewer,
>
> We are currently unaware of the extent to which our responses have clarified your concerns, and willing to provide further clarifications to you.
>
> Regards,
> Authors

---

### Decision · Program_Chairs · 2023-01-20

**Decision:**

Accept: poster

**Justification For Why Not Higher Score:**

Why it can not have a higher score:

A higher score would require the paper to address the weaknesses mentioned above, and to demonstrate more clearly the novelty and significance of its contributions. Specifically, the paper could:

- Relax the monotonicity assumption or discuss how Q-GenX can be extended or adapted to non-monotone cases, or provide more motivation and examples for the monotone setting.
- Provide a more detailed and rigorous comparison with existing methods for different slices of the VI problem, and highlight the trade-offs and benefits of Q-GenX in terms of convergence rate, communication cost, and robustness to noise.
- Include some experiments in the main body that showcase the performance and scalability of Q-GenX on realistic and challenging VI problems, such as GANs, and compare with state-of-the-art methods.
- Discuss or compare with some closely related works that use compression on top of distributed algorithms for solving VI problems, such as [1,2], and explain how Q-GenX differs or improves upon them.


**Justification For Why Not Lower Score:**


Why it can not have a lower score:

A lower score would not be justified, as the paper still presents a solid and well-organized work that proposes a practical and unified framework for monotone VI optimization, and provides strong theoretical guarantees for its convergence. The paper also surveys a large body of related works, and uses unbiased and adaptive compression techniques that are novel and efficient. The paper has potential to make an impact on the field of distributed optimization, and to stimulate further research on VI problems.

**Metareview: Summary, Strengths And Weaknesses:**

The three reviews are generally positive about the paper, but they also point out some weaknesses and areas for improvement. The main strengths of the paper are:

- It proposes a quantized, general extra-gradient framework (Q-GenX) that unifies previous methods for monotone variational inequality (VI) optimization, a general class of problems that includes convex minimization, saddle-point problems, and games.
- It provides strong theoretical guarantees for the convergence of Q-GenX under different assumptions on the noise in the stochastic gradients, and shows that increasing the number of processors can accelerate convergence.
- It uses unbiased and adaptive compression techniques for communicating local stochastic gradients, and an adaptive learning rate rule, which are practical and efficient.
- It is very well written and provides a comprehensive survey of related works.

The main weaknesses of the paper are:

- It only considers the monotone case, which is a restrictive assumption that does not hold for many applications of machine learning.
- It does not make clear how its contributions compare to existing methods for different slices of the VI problem, and what are the advantages and limitations of Q-GenX.
- It does not include any experiments in the main body, and the experiments in the appendix are minimal and incomplete.
- It does not discuss or compare with some closely related works that use compression on top of distributed algorithms for solving VI problems.



Conclusion:

Based on the reviews, the paper can be accepted for poster, as it meets the minimum criteria for acceptance, but it also has some room for improvement. The paper should address the reviewers' comments and suggestions in the final version, and provide more evidence and clarity for its contributions and comparisons.

References:

[1] Beznosikov, Aleksandr, Eduard Gorbunov, Hugo Berard, and Nicolas Loizou. "Stochastic gradient descent-ascent: Unified theory and new efficient methods." arXiv preprint arXiv:2202.07262 (2022).

[2] Beznosikov, Aleksandr, Dmitry Kovalev, Egor Shulgin, and Peter Richtarik. "Communication-efficient distributed optimization with quantized extra-gradient methods." arXiv preprint arXiv:2102.06171 (2021).

**Note From Pc:**

if the above contains the word "oral" or "spotlight" please see: "oral" presentation means -> notable-top-5% and "spotlight" means -> notable-top-25%. As stated in our emails, we are disassociating presentation type from AC recommendations